# LIGAND CONFORMATION GENERATION: FROM SINGLETON TO PAIRWISE

## ABSTRACT

Drug discovery is a time-consuming process, primarily due to the vast number of molecular structures that need to be explored. One of the challenges in drug design involves generating rational ligand conformations. For this task, most previous approaches fall into the singleton category, which solely rely on ligand molecular information to generate ligand conformations. In this work, we contend that the ligand-target interactions are also very important in providing crucial semantics for ligand generation. To address this, we introduce PsiDiff, a comprehensive diffusion model that incorporates target and ligand interactions, as well as ligand chemical properties. By transitioning from singleton to pairwise modeling, PsiDiff offers a more holistic approach. One challenge of the pairwise design is that the ligand-target binding site is not available in most cases and thus hinders the accurate message-passing between the ligand and target. To overcome this challenge, we employ graph prompt learning to bridge the gap between ligand and target graphs. The graph prompt learning of the insert patterns enables us to learn the hidden pairwise interaction at each diffusion step. Upon this, our model leverages the Target-Ligand Pairwise Graph Encoder (TLPE) and captures ligand prompt entity fusion and complex information. Experimental results demonstrate significant improvements in ligand conformation generation, with a remarkable 18% enhancement in Aligned RMSD compared to the baseline approach.

## 1 INTRODUCTION

The protracted nature of drug discovery stems primarily from the substantial search space it encompasses (Polishchuk et al., 2013; Du et al., 2022). In the realm of drug design, ligand conformations generation based on ligand molecular graphs is crucial for constructing low-energy molecules in 3D Euclidean space (Liu et al., 2023). Recent advancements in deep learning, especially generative models, have shown promise in efficiently selecting and ranking highly promising candidates for drug discovery, leading to significant time and cost savings (Dara et al., 2021; Stärk et al., 2022). Several notable contributions have emerged in this field. Shi* et al. (2020) introduce flow-based models, while Mansimov et al. (2019) present VAE-based models, for molecular coordinate generation. Additionally, Shi et al. (2021) and Xu et al. (2022) propose end-to-end models for estimating the gradient fields of atomic coordinates using denoising score matching and diffusion methods, respectively. Notably, GeoDiff incorporates a rot-translation invariant network design by employing a zero Center of Mass (CoM) system, utilizing rot-invariant inputs and a rot-equivariant projection.

However, all of the aforementioned methods generate ligand conformations in a singleton manner, which means their sole reliance on the ligand's molecular graph. This approach limits their capacity to consider additional factors or constraints beyond the ligand itself, leading to potential issues of producing invalid or incorrect conformations. For instance, as depicted in Figure 1(b)(c), GeoDiff overlooks the crucial ligand-target interaction, while target information encompasses essential chemical and geometric details, resulting in suboptimal drug molecular conformations. Consequently, these limitations may result in high-energy conformations that are unstable and not in equilibrium states.

Taking inspiration from Contrastive Language-Image Pre-training (CLIP) (Radford et al., 2021), who employ language to offer broader supervision, enhance generality, interpretability, and control in image synthesis, we leverage target information as an additional source of supervision. This method shifts from considering individual ligands in singleton to examining ligand-target interactions and

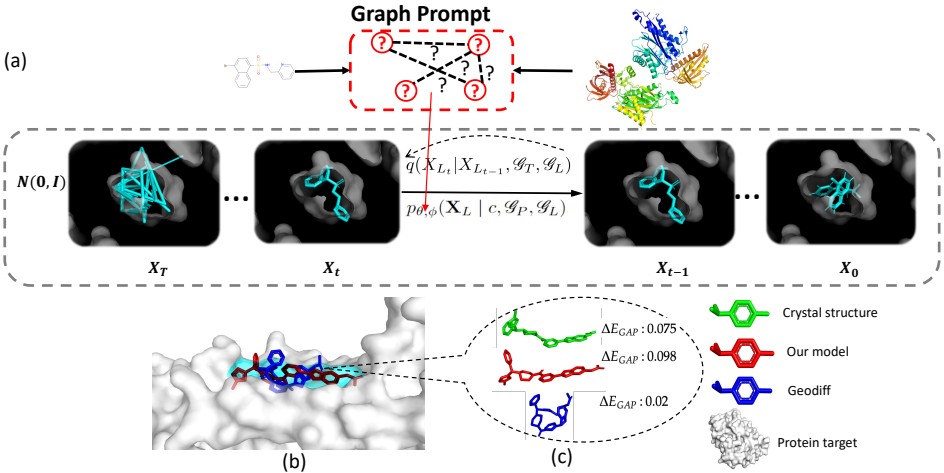

Figure 1: (a) Overview of our model: additional target and ligand pairwise interaction is incorporated into the diffusion model by graph prompt to generate ligand conformations. (b) Target information helps the model to capture the correct shapes of the 6cf7 ligand conformation structure, whereas (c) shows zoom-in pictures of the 6cf7 ligand. In (b) and (c), Green: reference ligand conformation crystal structure; Cyan: target pocket; Blue: wrong ligand conformation generated by GeoDiff-PDBBind2020, which fails to catch the extra long-range non-covalent interaction; Red: improved ligand conformation generated by our model, closer to the reference structure. $\Delta E_{GAP}$: HOMO-LUMO energy gap to describe the stability of conformations, the lower, the more stable.

relationships in pairs. This transition acknowledges the significance of pairwise interactions in shaping the behavior and properties of ligand-target pairs, enabling a more comprehensive understanding of the ligand conformation structure by considering the interactions between ligand-target pairs. Our proposed model, named Pairwise Structure Interaction Conditional Diffusion (PsiDiff), leverages ligand-target interaction information to guide the sampling process, while incorporating ligand chemical properties as additional constraints. By incorporating pairwise interactions, our model effectively addresses the challenges of semantic relevance. It takes into account the chemical and geometric features of the target protein, ligand-target complex, and local ligand chemical properties. This comprehensive approach ensures the generation of ligand conformations that possess meaningful context for drug design and selection.

In pairwise designs, one of the challenges is the difficulty in obtaining accurate ligand-target message passing due to the lack of ligand-target binding sites in most cases. To address this issue, PsiDiff utilizes graph prompts (Sun et al., 2023; 2022; Fang et al., 2022) to bridge the gap between ligand and target graphs. Graph prompts are inspired by natural language processing (NLP) prompts and are used to guide and improve the performance of graph models by providing structured input or instructions. In our model, graph prompts implicitly incorporate ligand-target interactions into the ligand conformation generation task. The prompt tokens are initialized with the structure of the target graph. The ligand-prompt message passing block (LPMP) and ligand-prompt complex graph insert the prompts into the ligand graphs hierarchically and throughout the diffusion steps. By incorporating target and ligand pairwise interactions through graph prompts, PsiDiff enhances stability and enables the generation of desirable ligand conformations. In summary, our main contributions are as follows:

- We introduce PsiDiff, a comprehensive diffusion model that incorporates ligand-target interactions. By transitioning from singleton to pairwise modeling, PsiDiff generates ligand conformations with meaningful biological semantics, significantly enhancing their relevance and usefulness in drug design.

- PsiDiff applies the concept of graph prompts to implicitly extract the pairwise ligand-target interaction and insert it into the ligand graph at each step of the diffusion model. This approach enables the generation of ligand conformations with pairwise information.

- The effectiveness of PsiDiff is demonstrated through experimental results on the PDBBind-2020 dataset. We observed significant improvements in Aligned RMSD compared to the baseline, achieving an enhancement of approximately 18%.

## 2 RELATED WORK

**Ligand-Target Docking Problem**   Ligand-target interaction (DTI) problems play a significant role in drug discovery by finding the suitable binding pose of ligand conformations onto some targets (McNutt et al., 2021; Halgren et al., 2004). In recent years, graph-based methods have emerged as a promising approach for addressing these problems. DiffSBDD (Schneuing et al., 2022) is a notable method that focuses on generating molecular structures specifically tailored for the docking problem by generating ligands nearby targets. It utilizes a diffusion-based approach to generate diverse molecular configurations centered around a given target molecule. EquiBind (Stärk et al., 2022) and TANKBind (Lu et al., 2022) are two docking methods that use graph neural networks to predict the coordinates of ligands and identify the binding pocket on the rigid protein. However, these methods are primarily focused on generating a single, optimal binding pose and may not capture the full conformational space of the ligand. Additionally, TANKBind requires further optimization from the ligand-target distance map to the ligand Euclidean coordinates. Furthermore, both DiffDock (Corso et al., 2023) and EquiBind require RDKit initialization at the beginning, which involves changing the atom positions by rotating and translating the entire molecule and rotating the torsion angles of the rotatable bonds. This initialization step can be problematic for molecules that cannot be initialized by RDKit (Riniker & Landrum, 2015) and limits the applicability of these methods to binding-pose conformation generation tasks (Du et al., 2022). In our method, the initialization is on the ligand atomic coordinates as Gaussian noise without any priorities. Moreover, instead of binding a molecule to some desired target, we use target information to improve the performance of molecular generation.

**Conditional generation**   Generation tasks that rely on self-information often involve predicting a predefined set of object categories. However, this form of supervision imposes significant limitations on the generality and usability of such models. These limitations arise from the fact that additional labeled data is necessary to effectively capture and predict visual concepts beyond the predefined categories (Radford et al., 2021). To address these limitations, Radford et al. (2021) introduces a conditioned generation paradigm that incorporates text information, enabling the model to leverage a broader source of supervision during the image generation process. The experimental results demonstrate that the inclusion of text-side information enhances the generality and usability of the generated images, leading to improved performance on image-generation tasks. Motivated by the success of this approach, we draw inspiration from Radford et al. (2021) and propose incorporating target information in the generation of ligand conformations. By considering target information alongside the ligand molecular graph, we aim to enhance the generality and usability of the generated ligand conformations, similar to the improvements observed in image generation tasks.

## 3 TARGET-LIGAND SIDE INFORMATION GUIDED DIFFUSIONS

**Problem Definition**   The problem at hand is defined as a *target and ligand pairwise interaction conditioning ligand conformation generation* task. Formally, the objective is to learn a parameterized distribution $p_{\theta,\phi}(\mathbf{X}_L \mid \mathscr{G}_P, \mathscr{G}_L, c)$ that approximates the Boltzmann distribution, which represents the probability distribution of ligand conformations coordinates $\mathbf{X}_L$ in the equilibrium states (Noé et al., 2018). Here the conditions for the generation task are target graphs $\mathscr{G}_P$ ligand graphs $\mathscr{G}_L$, and ligand chemical properties $c$. with detailed construction in Section 4.2. The learned distribution can then be utilized to independently draw heavy atom coordinates of ligands. In other words, given the target molecule graphs, ligand molecule graphs, and ligand chemical properties, our goal is to learn a probability distribution that generates conformations consistent with the given conditions.

**Forward Process**   Consider that the data distribution in the equilibrium states $q(\mathbf{X}_{L_0})$ undergoes a gradual transformation into a well-behaved and analytically tractable distribution $q(\mathbf{X}_{L_T})$, e.g. Normal distribution, through iterative applications of a Markov diffusion kernel $q(\mathbf{X}_{L_t} \mid \mathbf{X}_{L_{t-1}})$ for discrete time step from 1 to $T$,

$$q(\mathbf{X}_{L_t} \mid \mathbf{X}_{L_{t-1}}) = \mathcal{N}(\mathbf{X}_{L_t}; \sqrt{1 - \beta_t}\mathbf{X}_{L_{t-1}}, \beta_t \mathbf{I}) \tag{1}$$

where $\beta_1, ..., \beta_T$ is a fixed variance schedule at each time step, $\mathbf{X}_{L_t}$ denotes the ligand atom coordinates at step $t$. Note that the diffusion process above is discrete for t from 1 to $T$. If we take

continuous time steps by small time step change $\Delta t$, the forward process can be described by the Îto diffusion stochastic differential equation (SDE) (Anderson, 1982):

$$dX_L = f(X_L, t)dt + g(t)d\omega \tag{2}$$

where $\omega$ is a standard Wiener process, $f(X_L, t)$ is the drift coefficient calculated by $f(X_L, t) = -\frac{1}{2}\beta_t X_L$, and $g(t)$ is the diffusion coefficient derived by $g(t) = \sqrt{\beta_t}$. The detailed derivative of the Îto diffusion stochastic differential equation (SDE) from Equation 1 can be found in Appendix A.5.

**Reverse Process** Starting from $X_{L_T}$ drawn from some analytically tractable distribution $p_T(X_{L_T}) = q(X_{L_T})$, we are going to derive the data distribution $p_0$ and generate sample $X_{L_0}$ by reversing the diffusion process:

$$p(X_{L_{0:T-1}} \mid X_{L_T}) = \prod_{t=1}^{T} p(X_{L_{t-1}} \mid X_{L_t}) \tag{3}$$

To incorporate the chemical properties and target-ligand side information in the diffusion model, our key treatment lies in using neural networks to parameterize $p(X_{L_t} \mid X_{L_{t-1}})$ by $p_{\theta,\phi}(X_L \mid c, \mathscr{G}_P, \mathscr{G}_L)$. In particular, a new energy function is added to guide the generation process to respect the chemical properties. Following the Bayes' theorem whose details are in Appendix A.5, our new energy-guided SDE for the reverse process is:

$$dX_L = [f(X_L, t)dt - g(t)^2(s_\theta(X_{L_t}, \mathscr{G}_P, \mathscr{G}_L, t) - \lambda \nabla_{X_L} G_\phi(X_{L_t}, \mathscr{G}_L, t))dt] + g(t)\overline{\omega}_{X_L}, \tag{4}$$

where score function network $s_\theta(X_{L_t}, \mathscr{G}_P, \mathscr{G}_L, t)$ is the gradient of log-likelihood of the distribution at step $t$, i.e. $s_\theta(X_{L_t}, \mathscr{G}_P, \mathscr{G}_L, t) = \nabla_{\theta_{X_L}} \log p_{\theta,\phi}(X_L \mid c, \mathscr{G}_P, \mathscr{G}_L)$, $G_\phi$ is the energy function network designed to meet the chemical properties in guidance of the generation progress, $\lambda$ is the scalar weight on the guidance, and $\overline{\omega}_{X_L}$ is a standard Wiener process from $T$ to $0$. Using the Euler-Maruyama solver (Zhao et al., 2022; Song et al., 2021) to discretize the reverse SDE above, we get an iterative update equation of ligand conformation samples:

$$X_{L_{t-1}} = X_{L_t} - [f(X_{L_t}, t) - g(t)^2(s_\theta(X_{L_t}, \mathscr{G}_P, \mathscr{G}_L, t) - \lambda \nabla_{X_L} G_\phi(X_{L_t}, \mathscr{G}_L, t))] + g(t)z, \tag{5}$$

where $z \sim \mathcal{N}(0, 1)$. The sampling algorithm is in Appendix A.6.

**Learning Score and Energy Networks** From the reverse process above, the key networks we need to learn are the score function network $s_\theta(X_{L_t}, \mathscr{G}_P, \mathscr{G}_L, t)$ and the energy function network $G_\phi(X_{L_t}, \mathscr{G}_L, t)$. First of all, the following total training loss is adopted in this paper:

$$\mathcal{L}(\theta, \phi) = \mathcal{L}_s(\theta) + \lambda \mathcal{L}_G(\phi), \tag{6}$$

where $\mathcal{L}_s = \mathbb{E}[\|s_\theta - s\|^2]$ with $s$ sampled from the standard normal distribution and $\mathcal{L}_G = \mathbb{E}|G_\phi - c_{prop}|$ where $c_{prop}$ represent desired chemical properties. The training algorithm is given in Appendix A.6. In the next section, we are going to introduce the design detail of networks $s_\theta$ and $G_\phi$.

## 4 EQUIVARIANT TARGET-LIGAND NETWORK DESIGN

In this section, we begin by stating the principle of our model design, which is rot-translational invariance, as discussed in Section 4.1. We then delve into the detailed design of the parameterized score function $s_\theta(X_{L_t}, \mathscr{G}_P, \mathscr{G}_L, t)$ in Section 4.2. To involve ligand-target interaction, we apply a learnable graph prompt in the design of the score function. By incorporating pairwise information by graph prompt in $s_\theta$, which is overlooked in previous generation models, we address the problems illustrated in Figure 1. This pairwise design enables the model to consider the specific characteristics of the target pocket, leading to more reasonable generational of ligand conformations that align with ligand-target interaction. In addition, we explain the energy function $G_\phi(X_{L_t}, \mathscr{G}_L, t)$ in Section 4.3.

The energy model is parameterized by a pre-trained model $G_\phi$ based on the stacked Equivariant Graph Convolution Layer (EGCL) (Satorras et al., 2021; Hoogeboom et al., 2021) for chemical properties in 4.3.

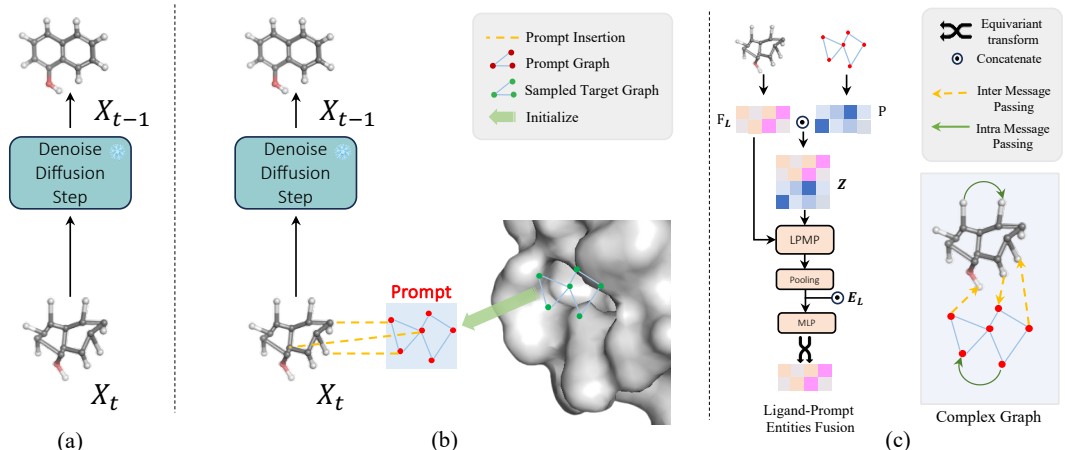

Figure 2: (a). Singleton methods consider ligand itself solely in diffusion method to generate ligand conformations. (b). Our method considers graph prompts to extract pairwise interaction and update ligand graph by the graph prompt to diffusion steps. (c). Two hierarchical insert patterns. In the left figure, ligand and prompt entity fusion insert pattern concatenate ligand and prompt graphs and fed it into LPMP together with ligand graph. In the right figure, we combine the two graphs as a complex graph and add edges for nodes within some Euclidean distance cutoff.

## 4.1 ROT-TRANSLATIONAL INVARIANCE

To ensure the score-based diffusion generation process maintains the desired rot-translational invariant property when working with 3D Euclidean coordinates, the generated distribution, denoted as $p_{\theta,\phi}(\mathbf{X}_{L_0})$, should remain unaffected by rotations and translations applied to the ligand coordinates. The rot-translational invariance on $p_{\theta,\phi}(\mathbf{X}_{L_0})$ can be guaranteed by the rot-translational invariance of $p_{\theta,\phi}(\mathbf{X}_{L_T})$ and the rot-translational invariance of markov kernal. Formally, we claim the following theorem:

**Theorem 1.** *If the initial density $p_{\theta,\phi}(\mathbf{X}_{L_T} \mid c, \mathscr{G}_P, \mathscr{G}_L)$ is rot-translational invariant, and the conditional Markov kernel $p_{\theta,\phi}(\mathbf{X}_{L_{t-1}} \mid \mathbf{X}_{L_t}, c, \mathscr{G}_P, \mathscr{G}_L)$ is rot-translational equivariant. Then generated density $p_{\theta,\phi}(\mathbf{X}_{L_0})$ is also rot-translational invariant. The rot-translational equivariance for the conditional Markov kernel is guaranteed by the rot-translational equivariance of the score function $\mathbf{s}_\theta$ and the energy function $G_\phi$.*

The invariance of $p_{\theta,\phi}(\mathbf{X}_{L_T})$ is achieved because the distribution represents an isotropic Gaussian, which is inherently invariant to rotations around the zero CoM (Xu et al., 2022). The zero CoM operation can again ensure the translational invariance for the Markov kernels. On the other hand, the rotational equivariance of the Markov kernel $p_{\theta,\phi}(\mathbf{X}_{L_{t-1}} \mid \mathbf{X}_{L_t})$ is accomplished by calculating the weighted average of invariant features depending on the 3D Euclidean coordinates through the utilization of atom pairwise distances, Gaussian curvature and mean curvature. We give the detailed proof for Theorem 1 in Appendix A.4. The parameterized $\mathbf{s}_\theta$ is the average of rot-equivariant transforms in graph neighborhood weighted by rot-invariant features based on such pairwise distances, Gaussian curvature and mean curvature.

## 4.2 TARGET-LIGAND PAIRWISE GRAPH PROMPT ENCODER (TLPE)

In this section, we present a detailed description of the parameterized encoder TLPE, which designed to approximating the score function $\mathbf{s}(\mathbf{X}_L)$. TLPE incorporats ligand-target interaction information and maintains the rot-translational equivariance of the Markov kernel. Our approach utilizes a graph prompt to learn ligand-target interaction during each diffusion step, as illustrated in Figure 2. We initialize graph prompt tokens based on target graphs, extract prompt tokens and token structures using the target graph feature extractor (Section 4.2), insert the graph prompt into the ligand graph (Section 4.2), and then use the updated ligand graph as input for the diffusion steps.

**Prompt Tokens and Token Structures** Ligand graphs are constructed from the molecular graphs denoted as $\mathscr{G}_L = (N_L, E_L)$. The nodes for the ligand molecular graph are heavy atoms with node features $\mathbf{F}_{L_j}$, while the edges denote the chemical covalent bonds. Ligand graphs are first fed into

graph neural networks to extract ligand node features $\mathbf{F}_L$ and edge features $\mathbf{E}_L$. The prompt graph is denoted as $\mathscr{G}_c = (P, S)$, where $P$ denotes the set of prompt tokens $p_i$ while $S$ denotes edges of the prompt graph. The number of tokens equals the number of down-sampled target graph nodes. We initialize the prompt token as the target graph constructed by the Surface Distance Function used in dMaSIF Sverrisson et al. (2021). Notably, targets are represented as point cloud graphs, where nodes correspond to point clouds in close proximity to the heavy atoms following dMaSIF (Sverrisson et al., 2021). The details for ligand and target graphs can be found in Appendix A.6.

**Inserting Patterns**  After constructing the graph prompt token, our next objective is to insert it into the ligand graph. We designed two inserting patterns to insert the graph prompt into the ligand graph hierarchically. *The first one* treats the ligand and prompt graphs as a new two-node graph, allowing messages to pass between them. This approach establishes effective communication between the two nodes by a feature assembling block called the Ligand-Prompt Message Passing Block (LPMP). This block facilitates the insertion of prompt graphs into ligands, enabling them to interact and exchange information. As shown in Figure 2(c) left, inspired by the message passing thought, we introduce the ligand prompt entity fusion block. The two nodes to be considered are $\mathbf{F}_L$ and $\mathbf{Z}$. Here $\mathbf{F}_L$ is the ligand node features and $\mathbf{Z} = \text{Concat}(\mathbf{F}_L, P)$ is the concatenated ligand-prompt node features.

To facilitate message passing between the newly constructed graph nodes, we employ five sub-blocks for layer-wise graph updates, with additional details provided in Appendix A.6. These sub-blocks cover all edges and iteratively produce $\mathbf{Z}$ over multiple layers, denoted as $\mathbf{Z} = \text{LPMP}(\mathbf{F}_L, \mathbf{Z})$. Our approach treats targets as fixed and rigid entities, focusing on updating the partitions within the ligand graphs. To transfer the concatenated node features to ligand nodes, we employ average pooling. Subsequently, we compute the output feature $\mathbf{F}_{L_{out_{local}}}$ by applying an MLP to the concatenation of ligand node and edge features, represented as $\mathbf{F}_{L_{out_{local}}} = \text{MLP}(\text{Pool}(\mathbf{Z}) \odot \mathbf{E}_{local})$.

*The second insertion pattern* involves creating a complex graph where nodes combine both ligand and prompt graph nodes. While the LPMP approach primarily focuses on ligand and prompt node feature interactions, emphasizing interactions between two complete entities. Here we aim to enhance the interpretation of inter-graph interactions at the edge level. To achieve this, we construct a complex graph that integrates ligand and prompt nodes. In this complex graph, we establish edges between nodes based on specific distance cutoffs. The edges connecting the ligand and prompt graphs represent ligand-prompt "inter-interactions," while edges within the ligand graphs account for long-range effects on non-covalent nodes. It is important to note that the prompt graph remains fixed throughout the diffusion process, and edges within the prompt are disregarded.

To build the feature extractor for the complex graph, we utilize SchNet (Schütt et al., 2017). This feature extractor enables message passing for the $l$-th layer as described in Eq. 7. In this context, $\Phi_{m_{global}}$ and $\Phi_{h_{global}}$ represent the parameterized complex branch network, while $\theta_{m_{global}}$ and $\theta_{h_{global}}$ correspond to the parameters within the complex branch.

$$\mathbf{m}_{C_{jy}} = \Phi_{m_{global}}(\mathbf{F}_{C_j^l}, \mathbf{F}_{C_y}^l, \mathbf{D}_{jy}, \mathbf{E}_{jy}; \theta_{m_{global}}), \mathbf{F}_{C_j}$$
$$= \Phi_{h_{global}}(\mathbf{F}_{C_j}^l, \sum_{y \in N(j)} \mathbf{m}_{C_{jy}}; \theta_{h_{global}}) \tag{7}$$

Where $y$ denotes the nodes in the ligand-prompt complex graph with $y \in \{ligand, prompt\}$, $j$ is the ligand node index. $m$ and $h$ denote the parameters for message passing and the aggregation on complex nodes, respectively. The output feature is then passed through an MLP together with complex edges $\mathbf{E}_{global}$ to get the output feature $\mathbf{F}_{L_{out_{global}}} = \text{MLP}(\mathbf{F}_C^L \odot \mathbf{E}_{global})$.

After that, we use the equivariant transform block to calculate the weighted average of the rot-translational invariant features. This block helps to transfer features with the same dimensionality as ligand edges to the dimensionality of ligand nodes. The details are shown in 4.2

**Rot-Translational Equivariance for TLPE**  We have two inserting patterns as discussed above. To make sure that $\mathbf{s}_\theta$ is rot-translational equivariant, both the two inserting patterns should satisfy rot-translational equivariance.

The rot-translational invariance of the LPMP block is satisfied because the two inputs for the LPMP block $\mathbf{F}_L, \mathbf{Z}$ are rot-translational invariant. They only depend on the invariant chemical features,

pairwise distances, and the Gaussian and mean curvature. Therefore, $\mathbf{F}_{L_{out_{local}}}$ is rot-translational invariant because $\mathbf{Z}$ and $\mathbf{E}_{local}$ only depend on the invariant chemical features, pairwise distances, and the Gaussian and mean curvature. As a result, the output of the LPMP block is also rot-translational invariant.

The rot-translational invariance of the complex inserting pattern satisfies since all the features are either dependent on pairwise distances or independent of coordinates. Overall $\mathbf{F}_{L_{out_{global}}}$ is rot-translational invariant because $\mathbf{Z}$ and $\mathbf{E}_{local}$ only depend on the invariant chemical features, pairwise distances, and the Gaussian and mean curvature. We provide the detailed proof in Appendix A.4.

We claim that if we requires the Markov Kernel being rot-translational equivariant, the score function should be rot-translational equivariant in Theorem 1. As discussed above, the output features for both insertting patterns are rot-translational invariant because all the features exhibit invariance since they are either dependent on pairwise distances or independent of coordinates. We have

$$\mathbf{s}_\theta = \sum_{j' \in N(j)} dir_{jj'} \mathbf{F}_{L_{out_{jj'}}}$$

as the equivariant transformation, where $dir_{jj'}$ denotes the unit director of the vector between the coordinates of two nodes, calculated as $dir_{jj'} = \frac{1}{D_{jj'}}(\mathbf{X}_{L_j} - \mathbf{X}_{L_{j'}})$. So the score $\mathbf{s}_\theta$ is the linear combination of roto-equivariant transforms $dir$ in graph neighborhood weighted by rot-invariant features $\mathbf{F}_{L_{out_{jj'}}}$. Here, $\mathbf{F}_{L_{out_{jj'}}}$ means $\mathbf{F}_{L_{out_{local}}}$ in ligand prompt entity fusion while $\mathbf{F}_{L_{out_{global}}}$ in the complex graph.

## 4.3 EQUIVARIANT ENERGY MODELS

The energy model utilized to guide the sampling process is formulated as the gradient of the estimation $G_\phi$. The energy model takes ligand molecular graphs as input, along with ligand atom coordinates. To train the model, we employ the stacked Equivariant Graph Convolution Layer (EGCL) (Satorras et al., 2021; Hoogeboom et al., 2021), with fixed ligand atom types. The Equivariant Graph Convolution Layer (EGCL) guarantees the transition equivariance by the zero-CoM operation. The model is rotational equivariant because there is only linear operation on the coordinates and all the nonlinear operations on coordinates-dependent functions using pairwise distance instead of coordinates. The details for the equivariant energy models are in Appendix A.6.

## 5 EXPERIMENTAL RESULTS

### 5.1 DATASET

We use PDBBind-2020 for both training and sampling in this work. Following the same data splitting strategy as Lu et al. (2022) and removing ligand-atom pairs with atoms outside the selected 32 atom types in Appendix A.6 or data that cannot be processed by Psi4 or RDKit for property calculation, we obtained 13,412, 1,172, and 337 pairs of complexes in the training, validation, and test sets, respectively. The test set does not contain any data that appear in or are similar to the training or validation sets.

Unlike traditional ligand conformation generation datasets such as GEOM (Ramakrishnan et al., 2014a), which contain no target data, PDBBind contains both ligand and target data, but they have a one-to-one correspondence. This enables us to effectively capture both intra-ligand long-range interactions and ligand-target 'inter-graph' interactions, as described in Section.4.2.

### 5.2 EXPERIMENT SETTING

We use Adam (Kingma & Ba, 2014) as the optimizer for both the diffusion and energy guidance models. The diffusion model was trained with 5000 steps for inference in the aligned RMSD experiment and 1000 steps for the RMSD experiments. It took around two days on eight Tesla A100 GPUs to train for 80 epochs.

During sampling, we add complex information only when $\sigma < 0.5$ for ligands with more than 50 atoms (i.e., large ligands) and when $\sigma < 3.4192$ for those with fewer than 50 atoms (i.e., small

ligands). For the pseudo-edge threshold, we used $8\mathring{A}$ as the intra-edge threshold and $2.8\mathring{A}$ as the inter-edge threshold. Empirically, atoms within $8\mathring{A}$ have non-covalent interactions inside a molecule. We chose the inter-edge threshold by first calculating the fraction of the number of atoms in the ligand and pocket, which was $7.08\%$. Then, we chose the $7.08\%$ quantile of the pairwise distances, which was $2.8\mathring{A}$. The experiment settings for the chemical property energy model are in Appendix A.6.

**Evaluation Metric**  We evaluate the generation quality in two aspects: similarity to the crystal conformations, which is evaluated by the aligned RMSD in Eq.8. For two conformations $\mathbf{X} \in \mathbb{R}^{n \times 3}$ and $\hat{\mathbf{X}} \in \mathbb{R}^{n \times 3}$, with $R_g$ denoting the rotation in SE(3) group, the alignment of two conformations can be evaluated by the Kabsch-aligned RMSD:

$$\text{RMSD}_{Align}(\mathbf{X}, \hat{\mathbf{X}}) = \min_{\mathbf{X}' \in R_g\hat{\mathbf{X}}} \text{RMSD}(\mathbf{X}, \mathbf{X}'), \tag{8}$$

where $RMSD(\mathbf{X}, \hat{\mathbf{X}}) = (\frac{1}{n}\sum_{j=1}^{n}\|\mathbf{X}_j, \hat{\mathbf{X}}_j\|^2)^{\frac{1}{2}}$.

## 5.3  RESULTS ON ALIGNED RMSD

In this section, we conduct a comparison by calculating the average of five generated conformations and evaluating them against baseline models, namely the ligand conformation generation method (GeoDiff (Xu et al., 2022)) and the docking method (TANKBind (Lu et al., 2022)). To ensure a fair evaluation, we employed the same training set as TANKBind (PDBBind-2020) and retrained the GeoDiff model on this dataset. It is worth noting that the performance of the original weights provided by GeoDiff, which were trained on GEOM-QM9 (Ramakrishnan et al., 2014b) and GEOM-Drugs (Axelrod & Gómez-Bombarelli, 2020) datasets, is even worse due to the disparity in data distribution between those datasets. More detailed results can be found in Appendix A.7. For clarity, we use the term GeoDiff-PDBBind to refer to the GeoDiff model retrained on the PDBBind dataset.

The quality of the generated conformations can be assessed using the aligned RMSD, as defined in Eq. 8. Table 1 presents the results. Notably, our method achieved a $17.7\%$ reduction in the median aligned RMSD compared to GeoDiff-PDBBind, and a $7.6\%$ reduction compared to TANKBind, without any additional optimization. Furthermore, by applying a simple force field optimization (Halgren, 1996), our method achieved a $20\%$ reduction compared to GeoDiff-PDBBind and a $10\%$ reduction compared to TANKBind. These improvements highlight the effectiveness of our approach in enhancing the quality of the generated conformations.

| Models | Aligned RMSD($\mathring{A}$)↓ | | | |
|---|---|---|---|---|
| | mean | 25th | 50th | 75th |
| GeoDiff-PDBBind | 2.79 | 1.61 | 2.47 | 3.58 |
| TANKBind | 2.61 | 1.43 | 2.20 | 3.15 |
| PsiDiff | 2.609 | 1.417 | 2.033 | 2.97 |
| PsiDiff + FF | **2.36** | **1.335** | **1.98** | **2.85** |

Table 1: RMSD after alignment by Kabsch algorithm on PDBBind-2020(filtered)

## 5.4  ABLATION STUDY FOR DIFFERENT STRUCTURES

To show the improvement in each of the new components we introduced in this paper, we conducted a comprehensive assessment of various factors, including the complex graph construction, the intra-ligand inside ligand and the inter-edge long-range connection between the ligand and target, the LPMP node feature assembler, and energy funtion guidance through ablation studies. By systematically analyzing the impact of these components, we gained valuable insights into their individual effects on the overall performance of the system.

As depicted in Figure 3, the blue conformation (without intra-ligand and inter-ligand-target long-range connections) exhibited a higher likelihood of instability and high energy due to the conformation collapsing together. However, the model trained without non-covalent edges did not converge successfully, as indicated in

| Models | Aligned RMSD($\mathring{A}$)↓ | | | |
|---|---|---|---|---|
| | mean | 25th | 50th | 75th |
| w/o complex branch | 2.72 | 1.63 | 2.17 | 3.07 |
| w/o LPMP | 2.73 | 1.52 | 2.17 | 3.35 |
| PsiDiff | **2.609** | **1.417** | **2.033** | **2.97** |

Table 2: Ablation study removing the new designed blocks will damage the performance.

Table 2. The absence of interaction edges between the ligand and target prevented the model from capturing crucial interactions, resulting in outliers in larger conformations and an unusually high mean value. This highlights the model's inability to accurately represent the system without considering non-covalent interactions.

For the yellow conformation (without ligand-target complex), reasonable poses, including the center position and orientation inside the pocket, were not consistently achieved. Although the aligned RMSD in Table 2 did not increase significantly due to the alignment and pose adjustments, the removal of the complex branch still impacted the performance of aligned RMSD.

The introduction of the LPMP feature assembler block enhanced the ligand's ability to capture the shape of the pocket by transferring chemical and geometric messages from the target nodes to the ligand nodes, as observed in the difference between the green and red conformations in Figure 3. The removal of the LPMP block, as shown in Table 2, adversely affected the performance of aligned RMSD.

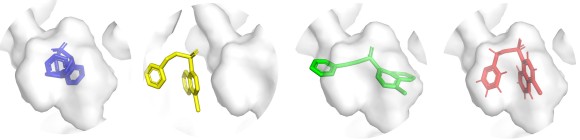

Figure 3: Ablation study for the effect of the intra-ligand long-range connection, the inter-edges connection between ligand and LPMP. The blue ligand is generated without long-range edges. The yellow ligand is the one without complex, the green one is the one without LPMP, and the red one is the standard version with all the components.

Furthermore, we compared the aligned RMSD for models utilizing different chemical properties as the energy function, in contrast to the model incorporating all three chemical properties used in our experiment. Each individual chemical property contributed to a slight decrease in RMSD, whereas employing all three properties yielded the best overall results. Further details and results are provided in Appendix A.7.

Our model can treat the DTI problem in an end-to-end manner without RDKit initialization. To evaluate the binding pose for the generated conformations, we used the ligand RMSD. We also compared our method to recent docking tasks as baselines to assess the performance of our approach in generating biologically meaningful conformations that are consistent with the given conditions while also being relevant for drug design and development. The detailed results are in the A.7.

## 6 CONCLUSION

This paper introduces PsiDiff, a conditional diffusion-based ligand conformation generation model that incorporates ligand-target interaction and chemical properties using an energy-guided score-based diffusion SDE. The model guarantees rot-translational equivariance through the zero-CoM system and equivariant transformation. The Target-Ligand Pairwise Graph Encoder (TLPE) employs the graph prompt idea to implicitly extract the unpreditable ligand-target interaction in each diffusion step. The graph prompt initializing by target graph is inserted to ligand graph. The insertion strategies consider the insertion hierarchically with ligand prompt entity fusion and complex graph. PsiDiff outperforms existing methods and holds promise for drug design and conformation generation tasks, with potential applications in protein-protein docking and ligand-protein soft docking projects.

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

# A APPENDIX

## A.1 CODE

All the code, data, and model checkpoints are available at https://anonymous.4open.science/r/PsiDiff-C441.

## A.2 NOTATION

We provide the main notations used in the paper here.

| Notations | |
|---|---|
| $\mathscr{G}_L$ | Ligand molecule graph |
| $\mathscr{G}_P$ | Target point cloud graph sampled similar to Sverrisson et al. (2021) |
| $\mathbf{X}_L \in \mathbb{R}^{n \times 3}, \mathbf{X}_P \in \mathbb{R}^{m \times 3}$ | Ligand and target coordinates |
| $C_L, center_P \in \mathbb{R}^3$ | Ligand and target center |
| $p_\theta(\mathbf{X}_L \mid \mathscr{G}_P, \mathscr{G}_L, c)$ | Parameterized ligand atom coordiantes distribution |
| $j, j'$ | Node index for ligand graphs |
| $i, i'$ | Node index for target graphs |
| $m, n$ | Number of nodes in target and ligand |
| $N_L, \mathbf{F}_L \in \mathbb{R}^{d_l \times n}$ | Ligand node and node features |
| $N_P, \mathbf{F}_P \in \mathbb{R}^{d_p \times m}$ | Target node and features |
| $N_C, \mathbf{F}_C \in \mathbb{R}^{d_p \times m}$ | Lig-Tar complex node and features |
| $\mathbf{Z} \in \mathbb{R}^{m \times n \times d}$ | Concat ligand and target feature |
| $\mathbf{D}_T, \mathbf{D}_L, \mathbf{D}_{inter}$ | Target, ligand, inter pairwise distances |
| $\mathbf{E}_{ii'}, \mathbf{E}_{jj'}, \mathbf{E}_{ij}$ | Target, ligand, inter edge features |
| $\mathbf{s}_\theta$ | Parameterized score funtion |
| $G_\phi$ | Energy Guidance model |
| $c$ | Chemical Properties |
| $\odot$ | Tensor concatenation |

Table 3: Notations used in the paper

## A.3 ROT-TRANSLATION INVARIANT

**Normalization** GeoDiff (Xu et al., 2022) operates on the ligand conformation's original coordinates, while the diffusion model operates on a normalized space (Sohl-Dickstein et al., 2015). To ensure consistency in the scalar values between small and large complexes, we initially normalize all the coordinates. Despite the normalization, our model remains rot-translation invariant due to the linearity of the transformation.

To apply the standard DDPM sampling process, we normalize both the ligand and target coordinates, matching their value range to that of the standard Gaussian noise in Equation 9.

$$\tilde{\mathbf{X}}_L = \frac{\mathbf{X}_L - center_P}{\sqrt{var_P}}, \tilde{\mathbf{X}}_P = \frac{\mathbf{X}_P - center_P}{\sqrt{var_P}} \tag{9}$$

$$\mathbf{X}_{L_0} = \tilde{\mathbf{X}}_{L_0} * \sqrt{var_P} + center_P \tag{10}$$

Here, $center_P$ is the center of mass for the target coordinates, $var_P$ is the maximum of the variance of the XYZ coordinates for the target, calculated as $var_P = max(var_{P_X}, var_{P_Y}, var_{P_Z})$. This normalization guarantees that the ligand and target coordinates share the same value range, which is crucial for the diffusion process.

Following the sampling process, we restore the generated conformations to their original coordinates using the recorded mean and variance, as illustrated in Equation 10. The targets are treated as fixed and rigid, with their centers and variances considered scalars. Consequently, the normalization transformations for the ligands retain rot-translational invariance. We provide detailed proofs of the rot-translational invariance with normalization in Appendix A.4.

**Rot-translational Invariant**   To ensure the score-based diffusion generation process maintains the desired rot-translational invariant property when working with 3D Euclidean coordinates, the generated distribution, denoted as $p_{\theta,\phi}(\mathbf{X}_{L_0})$, should remain unaffected by rotations and translations applied to the ligand coordinates as shown in . The invariance of $p_{\theta,\phi}(\mathbf{X}_{L_T})$ is achieved because the distribution represents an isotropic Gaussian, which is inherently invariant to rotations around the zero CoM (Xu et al., 2022). The zero CoM operation can again ensure the translational invariance for the Markov kernel. On the other hand, the rotational equivariance of the Markov kernel $p_{\theta,\phi}(\mathbf{X}_{L_{t-1}} \mid \mathbf{X}_{L_t})$ is accomplished by calculating the weighted average of invariant features depending on the 3D Euclidean coordinates through the utilization of atom pairwise distances, Gaussian curvature and mean curvature.

## A.4   Proof of Theorem. 1

If the initial density $p_{\theta,\phi}(\mathbf{X}_{L_T} \mid c, \mathscr{G}_P, \mathscr{G}_L)$ after normalization is rot-translational invariant, and the conditional Markov kernel $p_{\theta,\phi}(\mathbf{X}_{L_{t-1}} \mid \mathbf{X}_{L_t}, c, \mathscr{G}_P, \mathscr{G}_L)$ is rot-translational equivariant. Then generated density $p_{\theta,\phi}(\mathbf{X}_{L_0})$ is also rot-translational invariant. The rot-translational equivariance for the conditional Markov kernel is guaranteed by the rot-translational equivariance of the score function $\mathbf{s}_\theta$ and the energy function $G_\phi'$.

To prove the theorem, we first claim and prove the following lemmas.

**Lemma 2.** *If the initial density $p_{\theta,\phi}(\mathbf{X}_{L_T})$ after normalization is rotational invariant, and the morkov kernel $p_{\theta,\phi}(\mathbf{X}_{L_T} \mid c, \mathscr{G}_P, \mathscr{G}_L)$ is rotational equivariant. Then the final density $p_{\theta,\phi}(\mathbf{X}_{L_0})$ is also rotational invariant.*

*Proof.* Let $R_g$ denotes the rotation operation, we get:

$$p_{\theta,\phi}(R_g(\mathbf{X}_{L_0})) = \int p_{\theta,\phi}(R_g(\mathbf{X}_{L_T})p_{\theta,\phi}(R_g(\mathbf{X}_{L_0:T-1}) \mid R_g(\mathbf{X}_{L_T})))d\mathbf{x}_{1:\mathbf{T}} \tag{11}$$

$$= \int p(R_g(\mathbf{X}_{L_T}) \prod_{t=1}^{T} p_{\theta,\phi}(R_g(\mathbf{X}_{L_{t-1}}) \mid R_g(\mathbf{X}_{L_t})))d\mathbf{x}_{1:\mathbf{T}} \tag{12}$$

$$\tag{13}$$

The initial density $p_{\theta,\phi}(\mathbf{X}_{L_T})$ after normalization is rotational invariant, gives $p_{\theta,\phi}(\mathbf{X}_{L_T}) = p_{\theta,\phi}(R_g(\mathbf{X}_{L_T}))$
the morkov kernel is rotational invariant , gives $p_{\theta,\phi}(\mathbf{X}_{L_{t-1}} \mid \mathbf{X}_{L_t}) = (R_g(\mathbf{X}_{L_{t-1}}) \mid R_g(\mathbf{X}_{L_t}))$, then

$$p_{\theta,\phi}(R_g(\mathbf{X}_{L_0})) = \int p_{\theta,\phi}(\mathbf{X}_{L_T}) \prod_{t=1}^{T} p_{\theta,\phi}((\mathbf{X}_{L_{t-1}}) \mid \mathbf{X}_{L_t})d\mathbf{x}_{1:\mathbf{T}} \tag{14}$$

$$= \int p(\mathbf{X}_{L_T})p_{\theta,\phi}((\mathbf{X}_{L_{0:T-1}}) \mid \mathbf{X}_{L_T})d\mathbf{x}_{1:\mathbf{T}} \tag{15}$$

$$= p_{\theta,\phi}(\mathbf{X}_{L_0}) \tag{16}$$

$\square$

**Lemma 3.** *The noise vector fields $\mathbf{s}_\theta(\mathbf{X}_{L_T}, \mathscr{G}_P, \mathscr{G}_L, t)$ for the Markov kernels $p_{\theta,\phi}(\mathbf{X}_{L_T} \mid c, \mathscr{G}_P, \mathscr{G}_L)$ are rotational equivariant.*
*Formally, denote the ligand features and target features in ligand and target graphs as $\mathbf{F}_L$, $\mathbf{F}_T$, respectively,*

$$R_g\mathbf{s}_\theta(\mathbf{X}_{L_T}, \mathscr{G}_P, \mathscr{G}_L, t) = \mathbf{s}_\theta(R_g\mathbf{X}_{L_T}, R_g\mathscr{G}_P, R_g\mathscr{G}_L, t), \tag{17}$$

*where $R_g\mathbf{X}_{L_T}$ means take the rotation matrix $R_g$ on each ligand atom coordinates.*

*Proof.* In the ligand feature extractor, $\mathbf{F}_{L_j}$, $\mathbf{E}_{jj'}$ are rotational invariant because they do not depend on coordiantes. the distance $D_{jj'}$ is a scalar, which is also invariant,
so for Eq.  46 47:

$$R_g\mathbf{m}_{jj'} = \Phi_m(R_g\mathbf{F}_{L_j}, R_g\mathbf{F}_{L_{j'}}, R_gD_{jj'}, R_g\mathbf{E}_{jj'}; \theta_m) = \Phi_m(\mathbf{F}_{L_j}, \mathbf{F}_{L_{j'}}, D_{jj'}, \mathbf{E}_{jj'}; \theta_m) = \mathbf{m}_{jj'} \tag{18}$$

and

$$R_g \mathbf{F}_{L_j} = \Phi_h(R_g \mathbf{F}_{L_j}, \sum_{j' \in N(j)} R_g \mathbf{m}_{jj'}; \theta_h) = \Phi_h(\mathbf{F}_{L_j}, \sum_{j' \in N(j)} \mathbf{m}_{jj'}; \theta_h) = \mathbf{F}_{L_j} \qquad (19)$$

In the target feature extractor, $f_{chem_i}, f_{geom_i i'}$ are scalars, and also invariant, for for Eq. 49:

$$R_g \mathbf{F}_{P_i} = \Phi_p(R_g f_{chem_i}, R_g f_{geom_i i'}) = \Phi_p(f_{chem_i}, f_{geom_i i'}) = \mathbf{F}_{P_i} \qquad (20)$$

The feature assembler block only updates the node features, which are invariant,

$$R_g \mathbf{F}_C = \text{LTMP}(R_g \mathbf{F}_L, R_g \mathbf{Z}) = \text{LTMP}(\mathbf{F}_L, \mathbf{Z}) = \mathbf{F}_C \qquad (21)$$

where $\mathbf{Z} = \mathbf{F}_L \odot \mathbf{F}_P$, $\mathbf{E}$ is the edge features for the ligand-prompt complex, similar to the complex branch.

$$R_g \mathbf{F}_{L_{out}} = \text{AdaptiveAveragePool}(R_g \mathbf{F}_C) \odot R_g \mathbf{E} \qquad (22)$$
$$= \text{AdaptiveAveragePool}(\mathbf{F}_C) \odot \mathbf{E} \qquad (23)$$
$$= \mathbf{F}_{L_{out}} \qquad (24)$$

Finally, for the edge-to-node equivariant transformation

$$R_g \mathbf{X}_{L_j} = \sum_{j' \in N(j)} R_g \frac{1}{D_{jj'}}(R_g \mathbf{X}_{L_j} - R_g \mathbf{X}_{L_{j'}}) R_g \mathbf{F}_{L_{out_{jj'}}} \qquad (25)$$

$$= R_g \sum_{j' \in N(j)} dir_{jj'} \mathbf{F}_{L_{out_{jj'}}} \qquad (26)$$

$$= R_g \mathbf{X}_{L_j} \qquad (27)$$

Therefore Eq. 17 is satisfied. □

**Lemma 4.** *The energy model $G_\phi$ based on EGCL is rot-translational equivariant with zero-CoM.*

*Proof.* As the EGCL formulas shown in Eq. 45, the transition equivariance is satisfied by applying zero CoM. We show the rotation equivariance here.
The parameterized network $\Phi_w, \Phi_m, \Phi_x, \Phi_h, m_{jj'}$ is the message, $\mathbf{F}_{L_j}$ is the ligand node feature consisting of node types, time, and chemical properties, which is independent of coordinates and thus being rotational invariant.

$$m_{jj'} = \Phi_m(\mathbf{F}_{L_j}, \mathbf{F}_{L'_j}, \mathbf{D}^2_{jj'}, \mathbf{E}_{jj'}) = \Phi_m(\mathbf{F}_{L_j}, \mathbf{F}_{L'_j}, \|R_g \mathbf{X}_{L_j} - R_g \mathbf{X}_{L'_j}\|^2, \mathbf{E}_{jj'}) \qquad (28)$$

Where $\|R_g \mathbf{X}_{L_j} - R_g \mathbf{X}_{L'_j}\|^2 = (\mathbf{X}_{L_j} - \mathbf{X}_{L'_j})^T R_g^T R_g (\mathbf{X}_{L_j} - \mathbf{X}_{L'_j}) = \|\mathbf{X}_{L_j} - \mathbf{X}_{L'_j}\|^2 = \mathbf{D}^2_{jj'}$, so

$$m_{jj'} = \Phi_m(\mathbf{F}_{L_j}, \mathbf{F}_{L'_j}, \mathbf{D}^2_{jj'}, \mathbf{E}_{jj'}) \qquad (29)$$

Then,

$$\mathbf{F}_{L_j} = \Phi_\mathbf{h}(\mathbf{F}_{L_j}, \sum_{j \neq j'} w_{jj'} m_{jj'}), \qquad (30)$$

$$R_g \mathbf{X}_{L_j} = R_g \mathbf{X}_{L_j} + \sum_{j \neq j'} R_g \frac{\mathbf{X}_{L_j} - \mathbf{X}_{L'_j}}{\sqrt{\mathbf{D}^2_{jj'} + 1}} \Phi_x(\mathbf{F}_{L_j}, \mathbf{F}_{L'_j}, \mathbf{D}^2_{jj'}, \mathbf{E}_{jj'}) \qquad (31)$$

With rotation $R_g$, the model satisfies

$$R_g \mathbf{X}^{l+1}_{L_j}, \mathbf{F}^{l+1}_{L_j} = \text{EGCL}(R_g \mathbf{X}^l_{L_j}, R_g \mathbf{F}^l_{L_j})$$

The above equation means that if the coordiantes and features are all rotational equivariant on the EGCL layer $l$, then they are also rotational equivariant on next EGCL layer $l+1$. Then, the energy model $G_\phi$ is also rotational invariant. □

**Proof of Theorem. 1**

*Proof.* As discussed in GeoDiff Appendix A.5, the zero CoM operation can ensure the translational invariance for $p_{\theta,\phi}(\mathbf{X}_{L_0})$. The thing remaining to prove is the rotational invariance.
Then from Lemma 4, $G_\phi(\mathscr{G}_L, \mathbf{X}_L, c, t)$ is rotational invariant giving that

$$G_\phi(R_g\mathbf{X}_L, \mathbf{F}_{L_j}, c, t) = G_\phi(\mathbf{X}_L, \mathbf{F}_{L_j}, c, t)$$

Take derivatives and multiply $R_g$ on both sides,

$$\nabla_{\mathbf{X}_L} G_\phi(R_g\mathbf{X}_L, \mathbf{F}_{L_j}, c, t) = R_g \nabla_{R_g\mathbf{X}_L} G_\phi(\mathbf{X}_L, \mathbf{F}_{L_j}, c, t)$$

Then, together with Lemma 3, Equation 4 is also rotational equivariant. Then the markov kernel is rotational equivariant.

Then together with Lemma 2, $p_{\theta,\phi}(\mathbf{X}_{L_0})$ is also transnational invariant. Finally, with the help of CoM-free system, $p_{\theta,\phi}(\mathbf{X}_{L_0})$ is rot-translational invariant. $\square$

### A.5 MODEL FORMULATION DETAILS

**Forward Process** According to Sohl-Dickstein et al. (2015); Song et al. (2021), the data distribution in the equilibrium states $q(\mathbf{X}_{L_0})$ undergoes a gradual transformation into a well-behaved and analytically tractable distribution $q(\mathbf{X}_{L_T})$ through iterative application of a Markov diffusion kernel $q(\mathbf{X}_{L_t} \mid \mathbf{X}_{L_{t-1}})$ for discrete time step from 0 to $T$, where $\beta_1, ..., \beta_T$ is a fixed variance schedule at each time step.

$$q(\mathbf{X}_{L_t} \mid \mathbf{X}_{L_{t-1}}) = \mathcal{N}(\mathbf{X}_{L_t}; \sqrt{1-\beta_t}\mathbf{X}_{L_{t-1}}, \beta_t\mathbf{I}) \tag{32}$$

$$q(\mathbf{X}_{L_{1:T}} \mid \mathbf{X}_{L_0}) = \prod_{t=1}^{T} q(\mathbf{X}_{L_t} \mid \mathbf{X}_{L_{t-1}}) \tag{33}$$

Equivalently, 1 can be written as following with $\mathbf{z}_{t-1}$ being the standard Gaussian noise:

$$\mathbf{X}_{L_t} = \sqrt{1-\beta_t}\mathbf{X}_{L_{t-1}} + \sqrt{\beta_t}\mathbf{z}_{t-1}, t = 1, ..., T \tag{34}$$

According to Yang et al. (2022), to simplify the representation of $q(\mathbf{X}_{L_{1:T}} \mid \mathbf{X}_{L_0})$, let $\alpha_t = 1 - \beta_t$ and $\bar{\alpha}_t = \prod_{s=1}^{t} \alpha_s$, then:

$$q(\mathbf{X}_{L_{1:T}} \mid \mathbf{X}_{L_0}) = \mathcal{N}(\mathbf{X}_{L_t}; \sqrt{\bar{\alpha}_t}\mathbf{X}_{L_0}, (1-\bar{\alpha}_t)\mathbf{I}) \tag{35}$$

Equivalently,

$$\mathbf{X}_{L_t} = \sqrt{\bar{\alpha}_t}\mathbf{X}_{L_0} + \sqrt{1-\bar{\alpha}_t}\mathbf{s} \tag{36}$$

The above diffusion process is discrete from 0 to $T$. If we take continuous time steps by small time step change $\Delta t$, the forward process can be described by the Îto diffusion stochastic differential equation (SDE) (Anderson, 1982):

$$d\mathbf{X}_L = f(\mathbf{X}_L, t)dt + g(t)d\omega \tag{37}$$

where $\omega$ is a standard Wiener process, $f(\mathbf{X}_L, t)$ is the drift coefficient calculated by $f(\mathbf{X}_L, t) = -\frac{1}{2}\beta_t\mathbf{X}_L$, and $g(t)$ is the diffusion coefficient derived by $g(t) = \sqrt{\beta_t}$.

*Proof.*

$$\mathbf{X}_{L_t} = \sqrt{1-\beta_t}\mathbf{X}_{L_{t-1}} + \sqrt{\beta_t}\mathbf{z}_{t-1}, t = 1, ..., T \tag{38}$$

Define $\bar{\beta}_t = T\beta_t$, then 38 can be rewrite as:

$$\mathbf{X}_{L_t} = \sqrt{1 - \frac{\bar{\beta}_t}{T}}\mathbf{X}_{L_{t-1}} + \sqrt{\frac{\bar{\beta}_t}{T}}\mathbf{z}_{t-1}, t = 1, ..., T$$

take $\beta(\frac{t}{T}) = \bar{\beta}_t$, $\mathbf{X}_L(\frac{t}{T}) = \mathbf{X}_{L_t}$, $\mathbf{z}(\frac{t}{T}) = \mathbf{z}_t$, then for $t = \{0, \frac{1}{T}, ..., \frac{T-1}{T}\}$, $\Delta t = \frac{1}{T}$, we have:

$$\mathbf{X}_L(t + \Delta t) = \sqrt{1 - \beta(t + \Delta t)\Delta t}\mathbf{X}_L(t) + \sqrt{\beta(t + \Delta t)\Delta t}\mathbf{z}(t)$$

when $\Delta t \to 0$, $\sqrt{1 - \beta(t + \Delta t)\Delta t} = 1 - \frac{1}{2}\beta(t + \Delta t)\Delta t$, then:

$$\mathbf{X}_L(t + \Delta t) = \mathbf{X}_L(t) - \frac{1}{2}\beta(t + \Delta t)\Delta t\mathbf{X}_L(t) + \sqrt{\beta(t + \Delta t)\Delta t}\mathbf{z}(t)$$

then

$$d\mathbf{X}_L = f(\mathbf{X}_L, t)dt + g(t)d\omega$$

where $f(\mathbf{X}_L, t) = -\frac{1}{2}\beta_t\mathbf{X}_L$, $g(t) = \sqrt{(\beta_t)}$ □

**Reverse Process** According to Sohl-Dickstein et al. (2015); Song et al. (2021), starting from $\mathbf{X}_{L_T}$ drawn from some analytically tractable distribution $p_T$ and reversing the diffusion process, we can derive the data distribution $p_0$ and sample $\mathbf{X}_{L_0}$ from it. The reverse process can be described on the reverse-time SDE given by Sohl-Dickstein et al. (2015); Song et al. (2021):

$$d\mathbf{X}_L = [f(\mathbf{X}_L, t)\mathbf{X}_L dt - g(t)^2\mathbf{s}(\mathbf{X}_L, t)dt] + g(t)\overline{\omega}_{\mathbf{X}_L}, \quad (39)$$

where $\overline{\omega}_{\mathbf{X}_L}$ is a standard Wiener process from $T$ to $0$, $dt$ is a negative infinitesimal timestep, and score function $\mathbf{s}(\mathbf{X}_L, t)$ is the gradient of log-likelihood of the distribution at step t $\mathbf{s}(\mathbf{X}_L, t) = \nabla_{\mathbf{X}_L}\log p(\mathbf{X}_L)$, with $p(\mathbf{X}_L)$ being the marginal distribution of the SDE at time t.
Referring Song et al. (2021); Zhao et al. (2022), we can approximate the score function by some neural network $\mathbf{s}_\theta$ and thus get the MSE loss for scoring matching as follows:

$$\mathcal{L} = \mathbb{E}[\|\mathbf{s}_\theta - \mathbf{s}\|^2] \quad (40)$$

Specifically, we design the Target-Ligand Pairwise Graph Encoder (TLPE) in Section 4.2 to get the score function approximation.

To generate conformations, we need to solve the above reverse SDE. Song et al. (2021) utilize the Euler-Maruyama solver to discretize the reverse SDE iteratively:

$$\mathbf{X}_{L_{t-1}} = \mathbf{X}_{L_t} - [f(\mathbf{X}_{L_t}, t) - g(t)^2\mathbf{s}(\mathbf{X}_{L_t}, t)] + g(t)\mathbf{z}, \mathbf{z} \sim \mathcal{N}(0, 1) \quad (41)$$

To ensure that the score-based diffusion system applied to the ligand's Euclidean coordinates satisfies rot-translational invariance, GeoDiff employs the Center of Mass (CoM) system. This system removes the center of mass for the conformations at each step, guaranteeing translational invariance. For achieving rot-invariance, GeoDiff initially operates on edge features, which are scalar quantities and inherently rot-invariant. Subsequently, these features are projected into the coordinate system using an equivariant transformation. However, in addition to rot-translational invariance, certain local scalar chemical features such as Self-consistent field (SCF) energy, molecular orbital (HOMO)–lowest unoccupied molecular orbital (LUMO) energy gaps, and Marsili-Gasteiger Partial Charges also play a crucial role in equilibrium states but are not considered in GeoDiff.

Instead of focusing solely on the score function in GeoDiff $\nabla_{\mathbf{X}_L}\log p(\mathbf{X}_L \mid \mathscr{G}_L)$, we consider the controllable score function $\nabla_{\mathbf{X}_L}\log p(\mathbf{X}_L \mid \mathscr{G}_P, \mathscr{G}_L, c)$, where $c$ denotes the chemical properties mentioned above $\mathscr{G}_P, \mathscr{G}_L$ denotes the target graph and ligand graph, respectively. Here we define $p(\mathbf{X}_L \mid \mathscr{G}_P, \mathscr{G}_L)$ as the reverse process of q and is also a normal distribution while the mean and variance have no closed form. Then as mentioned in Song et al. (2021); Zhao et al. (2022), we apply Bayes' theorem $p(\mathbf{X}_L \mid c, \mathscr{G}_P, \mathscr{G}_L)p(c) = p(c|\mathbf{X}_L, \mathscr{G}_P, \mathscr{G}_L)p(\mathbf{X}_L|\mathscr{G}_P, \mathscr{G}_L)$ where $p(c)$ is independent to $X_L$. Here ligand chemical property and target graph are independent thus $p(c|\mathbf{X}_L, \mathscr{G}_P, \mathscr{G}_L) = p(c|\mathbf{X}_L, \mathscr{G}_L)$. Taking derivative with respect to $\mathbf{X}_L$ on both sides, it results in the controllable score function: resulting in the following controllable score function:

$$\nabla_{\mathbf{X}_L}\log p(\mathbf{X}_L \mid \mathscr{G}_P, \mathscr{G}_L, c) = \nabla_{\mathbf{X}_L}\log p(\mathbf{X}_L \mid \mathscr{G}_P, \mathscr{G}_L) + \nabla_{\mathbf{X}_L}\log p(c \mid \mathbf{X}_L, \mathscr{G}_L) \quad (42)$$

Then the reversed SDE controllable by the chemical properties can be described as follows:

$$d\mathbf{X}_L = [f(\mathbf{X}_L, c, t)dt - g(t)^2(\mathbf{s}(\mathbf{X}_L, \mathscr{G}_P, \mathscr{G}_L, t) - \lambda\nabla_{\mathbf{X}_L}G(\mathbf{X}_L, \mathscr{G}_L, t))dt] + g(t)\overline{\omega}_{\mathbf{X}_L}, \quad (43)$$

where $\overline{\omega}_{\mathbf{X}_L}$ is a standard Wiener process from $T$ to $0$, $dt$ is a negative infinitesimal timestep, and score function $\mathbf{s}(\mathbf{X}_L, \mathscr{G}_P, \mathscr{G}_L, t)$ is the gradient of log-likelihood of the distribution at step $t$, i.e. $\mathbf{s}(\mathbf{X}_L, \mathscr{G}_P, \mathscr{G}_L, t) = \nabla_{\mathbf{X}_L}\log p(\mathbf{X}_L \mid \mathscr{G}_P, \mathscr{G}_L)$, with $p(\mathbf{X}_L \mid \mathscr{G}_P, \mathscr{G}_L)$ being the marginal distribution of the SDE at time t. $\lambda$ is the scalar weight on the guidance, $G$ is the energy function for the three

chemical properties mentioned before. And the reversed SDE shown in 4 is called Energy-guided Reverse-time SDE (Zhao et al., 2022).

Similar to Zhao et al. (2022); Song et al. (2021), we utilize the Euler-Maruyama solver to discretize the reverse SDE and use the neural network to parameterize $p(\mathbf{X}_L \mid c, \mathcal{G}_P, \mathcal{G}_L)$ by $p_{\theta,\phi}(\mathbf{X}_L \mid c, \mathcal{G}_P, \mathcal{G}_L)$. Specifically, we parameterize $\mathbf{s}$ and $G$ by $\mathbf{s}_\theta$ and $G_\phi$, and then we get the updating of ligand conformation samples in each step:

$$\mathbf{X}_{L_{t-1}} = \mathbf{X}_{L_t} - [f(\mathbf{X}_{L_t}, t) - g(t)^2(\mathbf{s}_\theta(\mathbf{X}_{L_t}, \mathcal{G}_P, \mathcal{G}_L, t) - \lambda \nabla_{\mathbf{X}_L} G_\phi(\mathbf{X}_{L_t}, \mathcal{G}_L, t))] + g(t)\mathbf{z}, \mathbf{z} \sim \mathcal{N}(0, 1) \tag{44}$$

### A.6 MODEL DETAILS

**Hyperparameters**  The essential hyperparameters are shown in Table. 4.

Table 4: Search space for PsiDiff to perform well on the validation set. The best choices for hyperparameters are marked in **bold**.

| PARAMETERS | SEARCH SPACE |
|---|---|
| Atom Type Num (Protein) | 6, 28, **32** |
| Atom Type Num (Ligand) | **28** |
| Inter-edge Distance Cutoff | 2, 2.8, 5, 7, **8**, 10, 15 |
| Intra-edge Distance Cutoff | 2, **2.8**, 5, 7, 8, 10, 15 |
| Protein Downsampling Rate | 0.01, **0.03**, 0.05, 0.1, 1 |
| LTMP Depth | 1, **2**, 4, 6, 8 |
| Training complex loss rate | **1**, 0.8, 0.5, 0.4, 0.1, **0** |
| Learning Rate | **1e-3**, 1e-4, 1e-5 |
| Learning Rate Scheduler | Cosine annealing |
| Time steps | 1000, 5000 |

**Energy function in the energy guided diffusion model**  The energy model utilized to guide the sampling process is formulated as the gradient of the estimation $G_\phi$. The energy model takes ligand molecular graphs as input, along with ligand atom coordinates. To train the model, we employ the stacked Equivariant Graph Convolution Layer (EGCL) (Satorras et al., 2021; Hoogeboom et al., 2021), with fixed ligand atom types. Here, $G_\phi$ represents the parameterized predictions of the chemical properties by the guidance model. The Equivariant Graph Convolution Layer (EGCL) guarantees the transition equivariance by the zero-CoM operation. The model is rotational equivariant because there is only linear operation on the coordinates and all the nonlinear operations on coordinates-dependent functions using pairwise distance instead of coordinates as shown in 45.

$$m_{jj'} = \Psi_m(\mathbf{F}_{L_j}^l, \mathbf{F}_{L_j'}^l, \mathbf{D}_{jj'}^2, \mathbf{E}_{jj'}), w_{jj'} = \Psi_w m_{jj'}, \mathbf{F}_{L_j}^{l+1} = \Psi_\mathbf{h}(\mathbf{F}_{L_j}^l, \sum_{j \neq j'} w_{jj'} m_{jj'}),$$

$$\mathbf{X}_{L_j}^{l+1} = \mathbf{X}_{L_j}^l + \sum_{j \neq j'} \frac{\mathbf{X}_{L_j}^l - \mathbf{X}_{L_j'}^l}{\sqrt{\mathbf{D}_{jj'}^2 + 1}} \Psi_\mathbf{x}(\mathbf{F}_{L_j}^l, \mathbf{F}_{L_j'}^l, \mathbf{D}_{jj'}^2, \mathbf{E}_{jj'}) \tag{45}$$

Here, $\Psi_w, \Psi_m, \Psi_x, \Psi_h$ are learnable networks, $m_{jj'}$ is the message, $\mathbf{F}_{L_j}^l$ is the ligand node feature consisting of node types, time, and chemical properties. $\mathbf{E}_{jj'}$ is the edge feature, which is the chemical bond type. Both are independent of the coordinates and thus are rot-translational invariant. $\mathbf{D}_{jj'}$ is the Euclidean distance and thus also rot-translational invariant. Then the update for $\mathbf{X}_{L_j}^l$ is rot-translational equivariant.

**LTMP**  The LTMP feature assembler considers the ligand and complex graph as two nodes of a directed self-looped graph and tries to pass massages inside the graph. It consists of 5 sub-blocks: D to Z, Z to Z, Z to L, L to L, and L to Z. The detailed structures of these 5 blocks are shown in Figure 4.

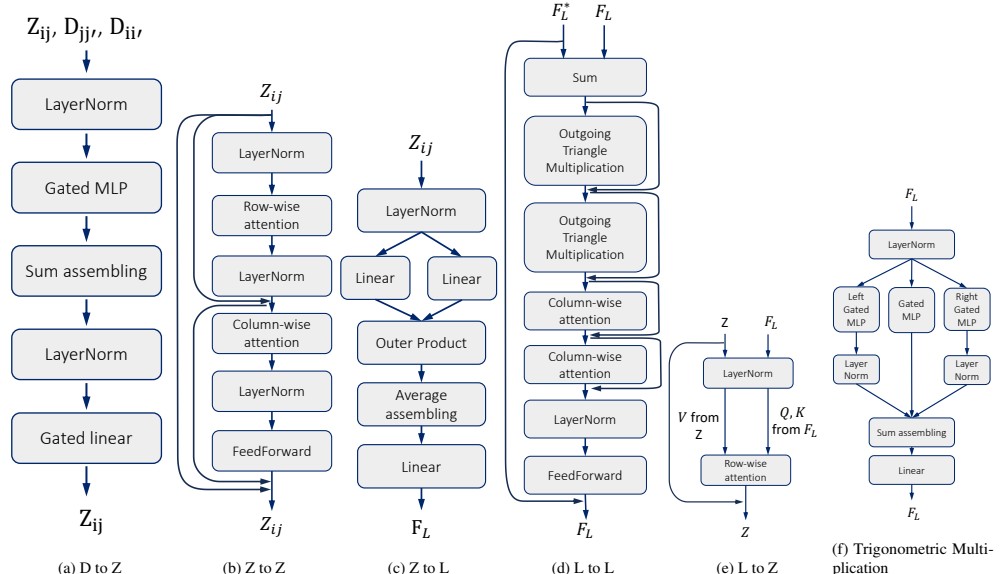

Figure 4: Sub-blocks of LTMP: Z to L, Z to Z, L to Z, D to Z, and L to L. The last subgraph shows the trigonometric multiplication block in the L to L sub-block.

**Graph representation and prompt inserting structure** Ligand graphs have nodes as the heavy atoms with node feature $F_L \in \mathbb{R}^{d_l \times n}$ and edges being the chemical covalent bonds with edge features $\mathbf{E_{jj'_{local}}}$. The node features are one-hot embedded from 28 atom types while the edge features are embedded by edge types and atomic pairwise distances. The atomic pairwise distances are rot-translational invariant and all other features are scalars not relative to coordinates, thus the ligand feature extractor is also rot-translational invariant.

In the provided ligand molecular graph, the consideration of strong chemical interactions solely through chemical bonds overlooks the potential long-range connections between nodes that lack covalent bonds but are in close proximity to each other in Euclidean space (Xu et al., 2022). This limitation disregards important interactions and relationships between such nodes. To overcome the limitations of previous approaches, we have integrated non-covalent interactions into our methodology similar to GeoDiff (Xu et al., 2022). Specifically, when the Euclidean distance between two ligand nodes is less than a designated threshold, we create pseudo edges between them. Additionally, the distance between these nodes is encoded as part of the edge features, allowing our approach to incorporate additional information about the spatial relationships between ligand nodes.
In our approach, we use a Graph Isomorphism Network (GIN) for the ligand-target interaction branch as the ligand feature extractor in equations 46 and 47. $\Phi_{m_{local}}$ and $\Phi_{h_{local}}$ denotes the parameterized ligand-target interaction networks. $\theta_{m_{local}}$ and $\theta_{h_{local}}$ denotes the parameters in the ligand-target interaction branch. As demonstrated in the equations below, all the features exhibit invariance since they are either dependent on pairwise distances or independent of coordinates.

$$\mathbf{m_{jj'}} = \Phi_{m_{local}}(\mathbf{F}_{L_j}^l, \mathbf{F}_{L_j'}^l, D_{jj'}, \mathbf{E}_{jj'}; \theta_{m_{local}}) \tag{46}$$

$$\mathbf{F}_{L_j}^{l+1} = \Phi_{h_{local}}(\mathbf{F}_{L_j}^l, \sum_{j' \in N(j)} \mathbf{m_{jj'}}; \theta_{h_{local}}) \tag{47}$$

Targets are represented as a point cloud graph, where the nodes correspond to point clouds in close proximity to the heavy atoms. The point clouds are sampled using the surface distance function (SDF) described in Equation 48. The motivation behind considering the SDF for sampling is rooted in the fact that the surface of the target predominantly influences its properties, and the SDF serves as a reliable representation of the protein surface (Zhu et al., 2010; Park et al., 2019; Venkatraman et al., 2009; Bordner & Gorin, 2007). Here, $\mathbf{a}_j$ denotes the protein atoms within the 32 atom types: (C, H, O, N, S, Se, Be, B, F, Mg, Si, P, Cl, V, Fe, Co, CU, Zn, As, Br, Ru, Rh, Sb, I, Re, Os, Ir, Pt, Hg, Ca, Na, Ni), $\mathbf{N_P}$ denotes the selected point clouds

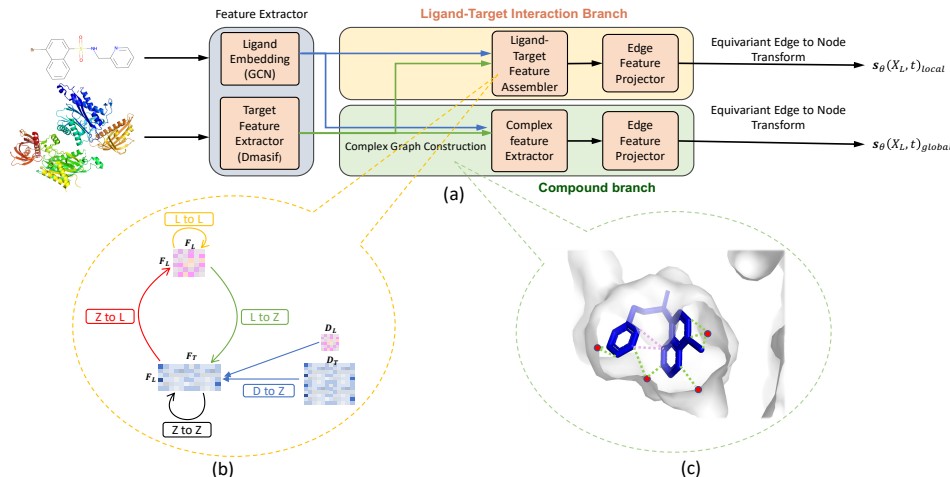

Figure 5: (a).TLPE: consists of the ligand-target interaction branch and complex branch; (b). Overview of LTMP block. (c). Ligand-target complex: red dots: protein surface nodes, green lines: inter-interactions between ligand and target graphs, pink lines: non-covalent pseudo-edges capturing long-range effects in ligand.

(nodes of the target graph), $\sigma$ is the experimental atom radius for $\mathbf{a}_j$, and w is the averaged atom radius.

$$\text{SDF}(\mathbf{N_P}) = -w \cdot \log \sum_{j=1}^{m} \exp(-\|\mathbf{N_P} - \mathbf{a}_j\|/\sigma) \tag{48}$$

The node features for the target graph encompass two main components: chemical features and geometric features. The chemical features consist of 32 node types, along with trainable chemical properties pertaining to the neighboring K atoms (K=16). Additionally, to capture the "shape" of the pocket surface more effectively, trainable geometric features, such as Gaussian curvatures and mean curvatures, are embedded within the node features. Formally, for the target point cloud graph, we follow the approach of Sverrisson et al. (2021) and extract the geometric and chemical features in equation 49. Here, $f^l_{chem_i}$ and $f_{geom^l_{ii'}}$ denote the chemical and geometric features for the target nodes, respectively.

$$\mathbf{F}^{l+1}_{P_i} = \Phi_p(f^l_{chem_i}, f^l_{geom_{ii'}}) \tag{49}$$

. Since Gaussian curvatures and mean curvatures are scalar quantities that remain invariant under rot-translation transformations, and the chemical properties are independent of the 3D Euclidean coordinates, the target feature extractor ensures rot-translational invariance. During the generation of ligand conformations, targets always remain unchanged and are regarded as rigid.

While our approach uses a trainable feature extractor dMaSIF (Sverrisson et al., 2021) to capture features of the target graphs represented by dense point clouds, using all the sampled points may derive more precise results on target features but also result in a computationally expensive feature assembler when passing massage between ligand and target. Therefore, dense target features may be redundant when the features are already extracted without much information loss. To address these issues, we use Fastest Point Sampling (FPS) (Ye et al., 2021; Nooruddin & Turk, 2003) to downsample the target point clouds after features are extracted. This downsampling after target feature extraction enables us to reduce the computational cost of the feature assembler while still preserving the relevant information needed for generating biologically meaningful conformations.

We try two combinations of backbone graph neural networks for the ligand feature extractor. The first one is Graph Convolution Network (GCN) for both ligand-target interaction and complex branches. The second one is SchNet (Schütt et al., 2017) for complex branch and Graph Isomorphism Network (GIN) for ligand-target interaction. The detailed structure is shown in Figure 6a. We also try a model similar to the energy model based on the EGNN model (Satorras et al., 2021; Hoogeboom et al., 2021) with the ligand atom types fixed and without the output MLP layer. The results show that the GCN version is better, so we finally it.

For the target graph, we choose the differentiable geodesic convolution-based surface point cloud

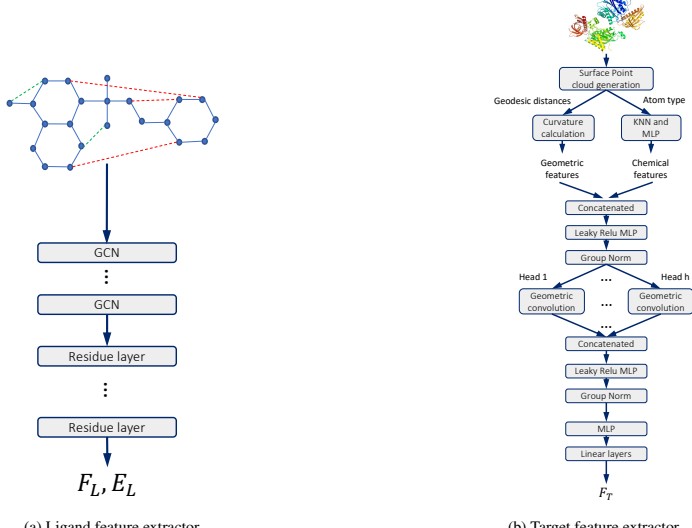

(a) Ligand feature extractor            (b) Target feature extractor

feature extractor dMaSIF, the detailed structure is shown in Figure 6b.

**Training and Sampling Algorithms** To ensure that the value ranges of the target and ligand node coordinates remain the same as the noises, which are sampled from standard normal distributions, we normalize the coordinates before taking gradient descent steps on the Epsilon network to train the noise score $\mathbf{s}_\theta$. The Pseudo code for training is shown in Algorithm.1.

The energy guidance is defined as the gradient of the L2 norm of the difference between predicted and reference chemical features. The training process for the energy guidance is shown in Algorithm. 1.

For the reverse process for sampling, we follow the standard DDPM algorithm with energy guidance on the chemical properties, as shown in Eq.4. After finishing all sampling steps, we transfer the coordinates value range back to the initial coordinates, as shown in the last line of Algorithm.2.

**Experiments settings** Three separate guidance models for gaps, energy, and charges were trained separately. Each model was trained on one A100(40GB) GPU for five days for 5000 epochs. The learning rate was set to be $2e^{-4}$ with a weight decay of $1e^{-16}$. We calculated the Self-consistent field (SCF) energy and molecular orbital (HOMO)–lowest unoccupied molecular orbital (LUMO) energy gaps using the Psi4 software (Parrish et al., 2017) and the Marsili-Gasteiger Partial Charges using RDKit (Riniker & Landrum, 2015).

## A.7 MORE RESULTS

**GeoDiff Pretrained Model on GEOM-QM9** GeoDiff is trained on GEOM-QM9 (Ramakrishnan et al., 2014b) and GEOM-Drugs (Axelrod & Gómez-Bombarelli, 2020) datasets, without any protein data inside them. Our model requires target information thus the above datasets are not available. We test the model weights given by GeoDiff and also retrain it on the PDBBind-2020 dataset. The direct testing on the given weights does not convergent for most of the ligands in the PDBBind datasets.

**Application on Ligand-Target-Interaction Problem** As shown in 5, without any extra optimization, our model achieves comparable results compared to the traditional method (GNINA (McNutt et al., 2021) and GLIDE (c.) (Halgren et al., 2004) and the deep learning method (EquiBind (Stärk et al., 2022) and TankBind (Lu et al., 2022)). With a simple one-step empirical force field (FF) (Halgren, 1996) optimization, our method outperforms most of the existing methods or their combination

---

**Algorithm 1** Generation Model Training

---

**Input:** $\mathscr{G}_L, \mathscr{G}_P, \mathbf{X}_{L_t}, c, \mathbf{X}_P, T$

1: **repeat**
2:      $t \sim Uniform(1, ..., T)$
3:      $\mathbf{X}_{L_0} \sim q(\mathbf{X}_{L_0})$
4:      $\tilde{\mathbf{X}}_{L_0} = \frac{\mathbf{X}_{L_0} - center_P}{\sqrt{var_P}}$                  ▷ Normalize ligand coordinates as Eq. 9
5:      $\tilde{\mathbf{X}}_P = \frac{\mathbf{X}_P - center_P}{\sqrt{var_P}}$                   ▷ Normalize target coordinates as Eq. 9
6:      $\mathbf{z} \sim \mathcal{N}(0, \mathbf{I})$
7:      $\tilde{\mathbf{X}}_{L_t} = \sqrt{\bar{\alpha}_t}\tilde{\mathbf{X}}_{L_0} + \sqrt{1 - \bar{\alpha}_t}\mathbf{z}$           ▷ Perturb ligand coordinates as Eq. 5
8:      Calculate $\mathbf{s}_\theta(\mathscr{G}_L, \mathscr{G}_P, \tilde{\mathbf{X}}_{L_t}, \tilde{\mathbf{X}}_P, c, t)$
9:      Sample $\mathbf{s}$ from the isotropic normal distribution
10:     Calculate $\mathcal{L}_{\mathbf{s}}$
11:     Take gradient descent step on $\nabla_\theta \mathcal{L}_{\mathbf{s}}$                 ▷ Loss function
12: **until** converged
13: **repeat**
14:     $t \sim Uniform(1, ..., T)$
15:     $\mathbf{X}_{L_0} \sim q(\mathbf{X}_{L_0})$
16:     $\tilde{\mathbf{X}}_{L_0} = \frac{\mathbf{X}_{L_0} - center_P}{\sqrt{var_P}}$                ▷ Normalize ligand coordinates as Eq. 9
17:     $\tilde{\mathbf{X}}_P = \frac{\mathbf{X}_P - center_P}{\sqrt{var_P}}$                 ▷ Normalize target coordinates as Eq. 9
18:     $\mathbf{z} \sim \mathcal{N}(0, \mathbf{I})$
19:     $\tilde{\mathbf{X}}_{L_t} = \sqrt{\bar{\alpha}_t}\tilde{\mathbf{X}}_{L_0} + \sqrt{1 - \bar{\alpha}_t}\mathbf{z}$         ▷ Perturb ligand coordinates as Eq. 5
20:     Calculate $G_\phi(\mathscr{G}_L, \mathbf{X}_L, c, t)$               ▷ Predict chemical features
21:     Calculate $G$ by RDKit and Psi4 packages
22:     Calculate $\mathcal{L}_G$
23:     Take gradient descent step on $\nabla_\phi \mathcal{L}_G$
24: **until** converged

---

**Algorithm 2** Equivariant Sampling

---

**Input:** $\mathscr{G}_L, \mathscr{G}_P, \mathbf{X}_P, c$
**Output:** $\mathbf{X}_{L_0}$

1:      $\tilde{\mathbf{X}}_P = \frac{\mathbf{X}_P - center_P}{\sqrt{var_P}}$                  ▷ Normalize target coordinates
2:      $\tilde{\mathbf{X}}_{L_T} \sim \mathcal{N}(0, \mathbf{I})$                    ▷ Random initial ligand coordinates
3:      **for** $t = T, ..., 1$ **do**
4:         $\mathbf{z} \sim \mathcal{N}(0, \mathbf{I})$ if $t > 1$, else $\mathbf{z} = 0$
5:         Calculate $\mathbf{s}_\theta(\mathscr{G}_L, \mathscr{G}_P, \tilde{\mathbf{X}}_{L_t}, \tilde{\mathbf{X}}_P, c, t)$
6:         Calculate $\nabla_{\mathbf{X}_L} G_\phi(\mathbf{X}_{L_t}, t)$
7:         Update $\tilde{\mathbf{X}}_{L_{t-1}}$ by Equation 5
8:         $\tilde{\mathbf{X}}_{L_{t-1}} = \tilde{\mathbf{X}}_{L_{t-1}} - \text{Center}(\tilde{\mathbf{X}}_{L_{t-1}})$         ▷ Take CoM
9:      **end for**
10:     $\mathbf{X}_{L_0} = \tilde{\mathbf{X}}_{L_0} * \sqrt{var_P} + center_P$
11:                     ▷ Transfer the coordinates back to the initial value range
12: **return** $\mathbf{X}_{L_0}$

---

| Models | Ligand RMSD Percentiles(Å)↓ | | |
|---|---|---|---|
| | 25th | 50th | 75th |
| GNINA | 2.4 | 7.7 | 17.9 |
| GLIDE (c.) | 2.6 | 9.3 | 28.1 |
| EquiBind | 3.8 | 6.2 | 10.3 |
| TANKBind | 2.4 | 4.28 | 7.5 |
| P2RANK+GNINA | **1.7** | 5.5 | 15.9 |
| EQUIBIND+GNINA | 1.8 | 4.9 | 13 |
| *GeoDiff-PDBBind | 29.21 | 40.33 | 79.62 |
| PsiDiff | 5.49 | 7.29 | 9.50 |
| PsiDiff + FF | 1.8 | **2.49** | **3.40** |

Table 5: Ligand RMSD on PDBBind-2020(filtered), Geodiff does not consider the position of ligands during docking, and centered the results to the origin of the Cartesian coordinate system.

| Models | Aligned RMSD(Å)↓ | |
|---|---|---|
| | mean | median |
| w/o guidance | 2.65 | 2.08 |
| SCF energy guidance | 2.649 | 2.07 |
| HOMO-LUMO energy gap | 2.65 | 2.06 |
| Marsili-Gasteiger Partial Charge | 2.636 | 2.04 |
| all 3 properties | **2.609** | **2.033** |

Table 6: Ablation study for using different chemical properties as energy functions

of median and 75th quantile.

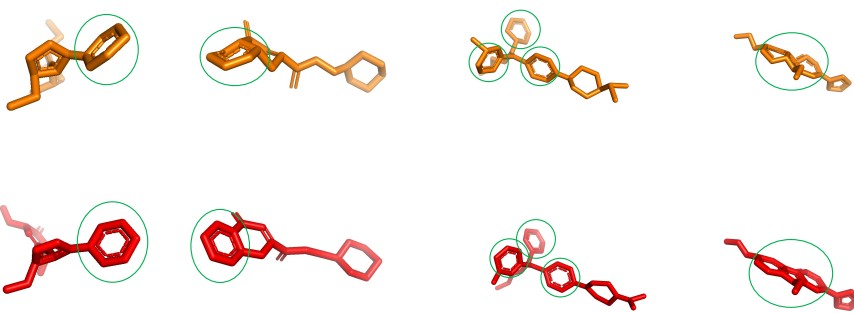

Figure 7: Ablation study for the effect of the guidance part. From left to right, the ligands are 5zjy, 5zk5, 6a6k, 6ggb. The red ligands are the ones with ligand property guidance while the orange ones are without guidance. The green circles point to the benzene rings in each ligand. Guidance helps to keep some geometric and chemical properties, such as the coplanarity of benzene rings.

**More Ablation Study**    While the improvement in aligned RMSD in Table 2 due to the guidance part may not be significant, further analysis revealed that guidance played a role in maintaining certain geometric and chemical properties, such as the coplanarity of benzene rings. These constraints assisted in generating more chemically reasonable molecules while satisfying energy or charge constraints. Although such local structure constraints might not drastically alter the overall structure, their presence explains the modest improvement in the aligned RMSD. Additional details and analysis can be found in Figure 7.

We do more ablation studies by using different chemical properties as energy functions. The results show that each chemical property helps to improve the performance a little. The best result is by using all the 3 chemical properties as shown in Table 6.

