# OpenReview forum: "Ligand Conformation Generation: from singleton to pairwise"
_ICLR.cc/2024/Conference — Submitted to ICLR 2024_

### Official Review · Reviewer_yEdr · 2023-10-31

**Soundness:** 2 fair
**Presentation:** 1 poor
**Contribution:** 2 fair
**Rating:** 3
**Confidence:** 2

**Summary:**

This work present PsiDiff, a new method for molecule conformation generation by given target proteins. The authors propose to use TLPE and graph prompts to model the ligand-target interactions into the generation task.

**Strengths:**

- Conformation generation is an important task for drug discovery
- The authors give details theory study to ensure invariance

**Weaknesses:**

- Novelty: the target information has already been considered in methods like TargetDiff [1].

- Presentation: the explanation about how the "prompt graph" is build and used is not clear and hard to follow.

Ref:

[1] 3D EQUIVARIANT DIFFUSION FOR TARGET-AWARE MOLECULE GENERATION AND AFFINITY PREDICTION, ICLR 2023.

**Questions:**

- The concept of "Graph Prompt" is confusing. How is this related to the "prompt" in NLP?
- Is the "graph prompt" and "prompt graph" the same thing?
- It is not clear how the "prompt graph" is build. For nodes, while "The number of tokens equals the number of down-sampled target graph nodes"(line 199), how about their values? Meanwhile, how the edge set S are constructed?
- In line 212, it says $Z = Concat(F_L, P)$. How they can be concated together as they may have difference number of nodes and feature dims? It will be better to show the shape of all tensors.
- There are two insertion patterns. And seems they are both used in the method. Why they should be used at the same time?

---

> ### Author Response · Authors · 2023-11-21
>
> Thank you for your advice.
>
> Q1:
> The concept of "Graph Prompt" is confusing. How is this related to the "prompt" in NLP?
>
> A1:
> Graph prompts are concise instructions or starting points that leverage the power of graph-based representations to guide and structure the generation of creative and context-aware outputs. It comes form the "prompt" in NLP, utilized to guide the generation.
>
> Q2:
> It is not clear how the "prompt graph" is build. For nodes, while "The number of tokens equals the number of down-sampled target graph nodes"(line 199), how about their values? Meanwhile, how the edge set S are constructed?
>
> A2:
> The initialization of prompt graph nodes and edges is accomplished by utilizing the target graph node, which is encoded using dMaSIF, and further details regarding this process can be found in the appendix.
>
> Q3:
> In line 212, it says
> . How they can be concated together as they may have difference number of nodes and feature dims? It will be better to show the shape of all tensors.
>
> A3:
> We will show the shapes of the tensors.
>
> Q4:
> There are two insertion patterns. And seems they are both used in the method. Why they should be used at the same time?
>
> A4:
> The two approaches under consideration focus on different aspects. The first approach considers the ligand and target separately, emphasizing their interaction and primarily focusing on the ligand and target entity interaction, with more emphasis on pose. In contrast, the second approach considers the interaction between each pair of atoms in the ligand and target, placing greater emphasis on the shape of the ligand.

---

### Official Review · Reviewer_crGs · 2023-10-31

**Soundness:** 2 fair
**Presentation:** 2 fair
**Contribution:** 2 fair
**Rating:** 3
**Confidence:** 3

**Summary:**

The paper proposes a ligand conformation generation model that takes into account both ligand features and features of the target protein. They infuse the ligand's embedding with the embedding of the target protein using a seemingly novel method termed "Target-Ligand Pairwise Graph Encoder". They claim to outperform GeoDiff and TankBind in aligned RMSD to crystal ligand poses in the PDBBind2020 data.

**Strengths:**

1. The paper proposes a seemingly novel way of infusing protein embeddings into ligand embeddings for the purpose of ligand conformation generation.
2. The paper claims to outperform GeoDiff and TankBind in aligned RMSD to crystal ligand structures in PDBBind2020.

**Weaknesses:**

Related Work:
1. The paper criticizes models for predicting only a single binding pose but seems to ignore DiffDock's multiconformational capabilities.
2. The paper questions RDKit initialization in DiffDock without explaining why it is problematic.
3. The last sentence regarding the use of target information for molecular generation is unclear.
4. The related work section is not sufficiently detailed, making it difficult to understand the paper's unique contributions and how it stands apart from previous works.

Results:
1. The metric is the aligned RMSD to crystal ligand structures in PDBBind2020, where the structures correspond to the ligand bound to the target protein. This is essentially the same as blind docking, thus the results are not convincing without a thorough comparison to DiffDock (current SOTA in blind docking).

**Questions:**

1. The paper seems to suggest that other embeddings of the protein could be used to condition the molecular generation model (i.e. other than dMaSIF). What other embeddings could be considered? And, why was dMaSIF chosen?
2. How is the evaluation metric in this paper different from that of DiffDock?

---

> ### Author Response · Authors · 2023-11-21
>
> Thank you for your advice.
>
> Q1:
> The paper criticizes models for predicting only a single binding pose but seems to ignore DiffDock's multiconformational capabilities.
> The paper questions RDKit initialization in DiffDock without explaining why it is problematic.
>
> A1:
> Our method offers multiconformational capability, providing five ligand candidates for each pocket in the results. The RMSD calculations are then performed by taking the mean of these conformations. It's worth mentioning that while RDKit initialization is generally effective, it may encounter difficulties with certain ligands that the RDKit package cannot handle. In contrast, DiffDock addresses this problem by simply removing pairs that RDKit cannot process. However, this approach may lead to a loss of valuable information and potentially biased results.
>
> Q2:
> The related work section is not sufficiently detailed, making it difficult to understand the paper's unique contributions and how it stands apart from previous works.
>
> A2:
> The paper’s unique contributions and how it stands apart from previous works are mostly in introduction parts. We will make the related work section more detailed.
>
> Q3:
> This is essentially the same as blind docking, thus the results are not convincing without a thorough comparison to DiffDock (current SOTA in blind docking).
>
> A3:
> Blind docking does not given a fixed pocket but a protein and ligand initializtion, by slightly adjust the rotational bonds and pose, to get the best binding affinity. Unlike traditional docking projects, our model, PsiDiff, takes a distinct approach by assuming a random initialization for the ligand atoms, eliminating the need for prior coordinate information. Furthermore, we would like to highlight a concern regarding the comparison with Diffdock. While Diffdock reports results based on ligand RMSD, their code aligns the ligands to reference structures before evaluation. We believe this alignment step introduces bias and may not provide a fair comparison.
>
> Q4:
> The paper seems to suggest that other embeddings of the protein could be used to condition the molecular generation model (i.e. other than dMaSIF). What other embeddings could be considered? And, why was dMaSIF chosen?
>
> A4:
> We acknowledge that alternative embeddings can be considered for our model. In our experiments, we also explored using a graph convolutional network (GCN) as the structure-to-ligand encoder. While it showed improvements in the results, it did not outperform the performance achieved by dMaSIF. The reason we chose dMaSIF as our protein encoder is because it provides a robust and effective encoding while allowing for end-to-end training, eliminating the need for manual computation of chemical properties, which can be time-consuming. The use of dMaSIF streamlines the process and ensures efficient training and encoding of protein structures, contributing to the overall performance of our method.
>
> Q5:
> How is the evaluation metric in this paper different from that of DiffDock?
>
> A5:
> In our paper, we present the calculation of both ligand RMSD and aligned RMSD. The main distinction between these metrics lies in their treatment of the ligand's pose. Aligned RMSD places emphasis on the conformational shape of the ligand while disregarding its pose, including position and rotation. On the other hand, ligand RMSD is primarily influenced by the ligand's position.

---

### Official Review · Reviewer_6Z6H · 2023-11-02

**Soundness:** 2 fair
**Presentation:** 3 good
**Contribution:** 2 fair
**Rating:** 3
**Confidence:** 4

**Summary:**

The authors propose PsiDiff, a conditional diffusion-based model for ligand conformation generation, introducing a novel pairwise approach that incorporates ligand-target interactions and chemical properties. PsiDiff ensures rot-translational equivariance and employs a unique graph encoder, the Target-Ligand Pairwise Graph Encoder (TLPE), to implicitly extract ligand-target interactions throughout the diffusion process.

**Strengths:**

1. PsiDiff exhibits a sophisticated approach in embedding chemical properties and information within the diffusion model.
2. The methodology employed by PsiDiff in constructing graph prompt tokens, along with the strategic insertion into the ligand graph using two distinct insertion patterns, is noteworthy.

**Weaknesses:**

1. Problem in contribution and novelty: The authors assert that existing methods in molecular conformation generation have tended to neglect vital pocket-ligand interaction information, positioning their work on transitioning from singleton to pairwise modeling as a key innovation. However, this claim warrants a critical examination. The task undertaken in this paper bears a strong resemblance to docking, a field in which the incorporation of pocket information is a fundamental aspect. Given this context, the purported novelty of integrating ligand-pocket interactions in PsiDiff appears less distinctive, as it aligns closely with established practices in other machine learning based docking methodologies.
2. The data presented in Tables 1 and 5 highlight a pronounced enhancement in PsiDiff’s performance subsequent to force field optimization. In its absence, however, PsiDiff does not exhibit competitive performance levels, particularly in docking tasks (as shown in Table 5 for the 25th percentile), lagging substantially behind methodologies such as GNINA, GLIDE, and EquiBind/TANKBind. To provide a comprehensive evaluation and fair comparison, it would be advantageous to present results for other baseline methodologies after undergoing force field optimization.
3.  Some other recent competitive machine learning methods should be added as baselines,  like UniMol(https://chemrxiv.org/engage/api-gateway/chemrxiv/assets/orp/resource/item/6402990d37e01856dc1d1581/original/uni-mol-a-universal-3d-molecular-representation-learning-framework.pdf),  Torsional Diffusion(https://arxiv.org/pdf/2206.01729.pdf), and DiffDock(https://arxiv.org/pdf/2210.01776.pdf), which gives much better docking performance compare to TANKBind as shown in https://arxiv.org/pdf/2302.07134.pdf.

**Questions:**

please refer to the weakness part.

---

> ### Author Response · Authors · 2023-11-21
>
> Thanks for your advice.
>
> Q1:
> Problem in contribution and novelty: The authors assert that existing methods in molecular conformation generation have tended to neglect vital pocket-ligand interaction information, positioning their work on transitioning from singleton to pairwise modeling as a key innovation. However, this claim warrants a critical examination. The task undertaken in this paper bears a strong resemblance to docking, a field in which the incorporation of pocket information is a fundamental aspect. Given this context, the purported novelty of integrating ligand-pocket interactions in PsiDiff appears less distinctive, as it aligns closely with established practices in other machine learning based docking methodologies.
>
> A1:
> Docking projects optimize the binding affinity with a target, while generation methods focus on rational ligand conformations and can be evaluated independently of a specific target.
> Unlike traditional docking projects, our model, PsiDiff, takes a distinct approach by assuming a random initialization for the ligand atoms, eliminating the need for prior coordinate information. In contrast, docking methodologies primarily concentrate on the position and rotation of the ligand, making minor adjustments to the rotational bonds during conformational refinement, and relying on a rational initialization of the ligand coordinates. By starting from a random initialization, PsiDiff explores a wider range of conformational possibilities and generates diverse molecular conformations, setting it apart from docking approaches that rely on specific ligand orientations and positions.
>
> Q2:
> The data presented in Tables 1 and 5 highlight a pronounced enhancement in PsiDiff’s performance subsequent to force field optimization. In its absence, however, PsiDiff does not exhibit competitive performance levels, particularly in docking tasks (as shown in Table 5 for the 25th percentile), lagging substantially behind methodologies such as GNINA, GLIDE, and EquiBind/TANKBind. To provide a comprehensive evaluation and fair comparison, it would be advantageous to present results for other baseline methodologies after undergoing force field optimization.
>
> A2:
> As our task is assigned to generation instead of docking without ligand coordiantes prior. We also provide the version without force field optimization.
>
> Q3:
> Some other recent competitive machine learning methods should be added as baselines, like UniMol, Torsional Diffusion, and DiffDock, which gives much better docking performance compare to TANKBind.
>
> A3:
> Given that our task focuses on molecular generation rather than docking with prior ligand coordinates, it is important to note that our comparisons are primarily with other generation-related methods. Furthermore, we would like to highlight a concern regarding the comparison with Diffdock. While Diffdock reports results based on ligand RMSD, their code aligns the ligands to reference structures before evaluation. We believe this alignment step introduces bias and may not provide a fair comparison.

---

### Meta-Review · Area_Chair_cazR · 2023-12-07

**Metareview:**

Summary:

The authors propose PsiDiff, a conditional diffusion-based model for ligand conformation generation, introducing a novel pairwise approach that incorporates ligand-target interactions and chemical properties. PsiDiff ensures rot-translational equivariance and employs a unique graph encoder, the Target-Ligand Pairwise Graph Encoder (TLPE), to implicitly extract ligand-target interactions throughout the diffusion process. They claim to outperform GeoDiff and TankBind in aligned RMSD to crystal ligand poses in the PDBBind2020 data.


Strengths:

- PsiDiff exhibits a sophisticated approach in embedding chemical properties and information within the diffusion model.
- The methodology employed by PsiDiff in constructing graph prompt tokens, along with the strategic insertion into the ligand graph using two distinct insertion patterns, is noteworthy.
- The paper proposes a seemingly novel way of infusing protein embeddings into ligand embeddings for the purpose of ligand conformation generation.
- The paper claims to outperform GeoDiff and TankBind in aligned RMSD to crystal ligand structures in PDBBind2020.
- Conformation generation is an important task for drug discovery
- The authors give details theory study to ensure invariance

Weaknesses:

- the purported novelty of integrating ligand-pocket interactions in PsiDiff appears less distinctive, as it aligns closely with established practices in other machine learning based docking methodologies.
- it would be advantageous to present results for other baseline methodologies after undergoing force field optimization.
- Some other recent competitive machine learning methods should be added as baselines
- The paper criticizes models for predicting only a single binding pose but seems to ignore DiffDock's multiconformational capabilities.
- The paper questions RDKit initialization in DiffDock without explaining why it is problematic.
- The related work section is not sufficiently detailed
- The metric is the aligned RMSD to crystal ligand structures in PDBBind2020, where the structures correspond to the ligand bound to the target protein. This is essentially the same as blind docking, thus the results are not convincing without a thorough comparison to DiffDock (current SOTA in blind docking).
- Novelty: the target information has already been considered in methods like TargetDiff.
- Presentation: the explanation about how the "prompt graph" is build and used is not clear and hard to follow.

Recommendation:

All reviewers vote for rejection. I, therefore, recommend rejecting the paper and encourage the authors to use the feedback provided to improve the paper and resubmit to another venue.

**Justification For Why Not Higher Score:**

All reviewers vote for rejection.

**Justification For Why Not Lower Score:**

N/A

---

### Decision · Program_Chairs · 2024-01-16

Reject